# What Factors Affect Multi-Modal In-Context Learning? An In-Depth Exploration

**Libo Qin**[‡][*]    **Qiguang Chen**[†][*]    **Hao Fei**[◇]    **Zhi Chen**[♣]    **Min Li**[‡]    **Wanxiang Che**[†]

[‡] School of Computer Science and Engineering, Central South University
[†] Research Center for Social Computing and Information Retrieval
[†] Harbin Institute of Technology    [◇] Tsinghua University    [♣] ByteDance
lbqin@csu.edu.cn, qgchen@ir.hit.edu.cn

## Abstract

Recently, rapid advancements in Multi-Modal In-Context Learning (MM-ICL) have achieved notable success, which is capable of achieving superior performance across various tasks without requiring additional parameter tuning. However, the underlying rules for the effectiveness of MM-ICL remain under-explored. To fill this gap, this work aims to investigate the research question: *"What factors affect the performance of MM-ICL?"* To this end, we investigate extensive experiments on the three core steps of MM-ICL including demonstration retrieval, demonstration ordering, and prompt construction using 6 vision large language models and 20 strategies. Our findings highlight (1) the necessity of a multi-modal retriever for demonstration retrieval, (2) the importance of intra-demonstration ordering over inter-demonstration ordering, and (3) the enhancement of task comprehension through introductory instructions in prompts. We hope this study can serve as a foundational guide for optimizing MM-ICL strategies in future research.

## 1 Introduction

Recently, Large Language Models (LLMs) have demonstrated remarkable advancements, showcasing proficiency in a wide range of tasks [Zhao et al., 2023a, Qin et al., 2023, 2024, Hu et al., 2023, Pan et al., 2023]. Notably, advanced LLMs exhibit the emergence of novel capabilities such as In-Context Learning (ICL) [Wei et al., 2022a, Dong et al., 2022, Zhuang et al., 2023], which optimize task performance by incorporating demonstrations into input prompts [Giannou et al., 2023, Li et al., 2023d, Wies et al., 2023, Zhou et al., 2022]. In particular, multi-modal in-context-learning (MM-ICL) is capable of utilizing multi-modal demonstrations to quickly adapt to the downstream task without parameter tuning [Yin et al., 2023, He et al., 2023, Zhang et al., 2024, Li and Lu, 2024].

In the literature, a series of works emerge to enhance MM-ICL. Specifically, Gong et al. [2023] manually create a general template with multiple images and corresponding responses during instruction-tuning (IT) stage to improve MM-ICL. Tsimpoukelli et al. [2021], Li et al. [2023b], Doveh et al. [2024] and Zhao et al. [2024] develop task-specific MM-ICL templates during the IT stage, further extending its capabilities across more domains. Li et al. [2023a] introduce OtterHD, adapting MM-ICL for high-definition image tasks. Furthermore, Sun et al. [2023] and Tian et al. [2024] explore the potential of MM-ICL in the image generation tasks. Jin et al. [2024] provide compelling evidence for the effectiveness of MM-ICL in comprehending game instructions. Zong et al. [2024] and Shukor et al. [2024] develop fine-grained benchmarks and evaluate the MM-ICL in classification tasks.

While significant progress has been witnessed in MM-ICL, the existing work still mainly focuses on how to optimize MM-ICL, ignoring the underlying factors that influence its effectiveness and

---

[*]Equal Contribution

38th Conference on Neural Information Processing Systems (NeurIPS 2024).

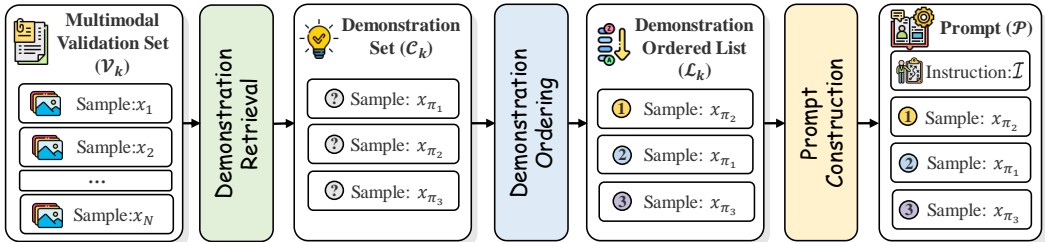

Figure 1: The whole process of prompting creation for multi-modal in-context-learning.

performance. Such gap impedes a comprehensive understanding of the mechanisms and performance determinants of MM-ICL, thereby limiting further exploration and research in this field. Motivated by this, this paper aims to systematically investigate the research question: *What factors affect the performance of MM-ICL?*, hoping to offer a unified view and guideline for researchers to build better MM-ICL. Specifically, as illustrated in Figure 1, the MM-ICL process comprises three steps: demonstration retrieval, demonstration ordering, and prompt construction. Therefore, We systematically investigate the following sub-questions: (a) *how to select multi-modal demonstrations* (Sec. 3.1); (b) *how to order multi-modal demonstrations* (Sec. 3.2); and (c) *how to construct MM-ICL prompts* (Sec. 3.3) to this end. To achieve this, we conduct detailed experiments on MM-ICL using 20 strategies across 4 tasks with 6 representative vision large language models (VLLMs).

Through extensive investigations, the main findings are as follows:

- **Multi-modal alignment is the bottleneck for MM-ICL.** Our analysis confirms that, on average, multi-modal retrieval methods outperform single-modal ones. Furthermore, multi-modal alignment in VLLMs has a greater impact on MM-ICL effectiveness than parameter size, identifying alignment as the key limitation in both backbone structure and demonstration quality.

- **Intra-demonstration ordering holds greater importance than inter-demonstration ordering.** Our investigation first indicates that the intra-demonstration ordering, particularly the ordering of modalities, greatly influences model performance more than demonstration arrangement.

- **Introductory instruction guides better task understanding for MM-ICL.** To construct a comprehensive MM-ICL prompt, it is essential to include introductory instructions preceding the demonstrations. This approach consistently enhances the performance of MM-ICL campared with summative instruction placed after demonstrations, and intra-demonstration instruction.

## 2 Background

In this work, we formally present the prompt building process for MM-ICL. As depicted in Figure 1, the process of prompt building for MM-ICL involves three sequential stages:

**(1) Demonstration Retrieval:** The core MM-ICL requires retrieval to obtain demonstrations that can help MM-ICL. Formally, given a validation dataset $\mathcal{V}_n = \{x_1, x_2, \ldots, x_n\}$, each multi-modal sample $x_i$ includes textual input $I_i^{txt}$, visual input $I_i^{vis}$, and output $O_i$. For a specific test query $q$, this step aims to identify a subset of relevant demonstrations $\mathcal{C}_k = \{x_{\pi_j}\}_{j=1}^k$, where $x_{\pi_j} \in \mathcal{V}_n$.

**(2) Demonstration Ordering:** Researches [Lu et al., 2022b, Wu et al., 2023, Xiang et al., 2024] show that LLMs are highly sensitive to the order of demonstrations. Thus, arranging these demonstrations effectively is crucial for MM-ICL. After retrieving relevant demonstrations, we must rearrange the sequence $\mathcal{L}_k = [x_{\sigma_j}]_{j=1}^k$, which will be used to construct the prompt.

**(3) Prompt Construction:** Previous research indicates that using delimiters and instructions can significantly enhance textual ICL capabilities [Min et al., 2022, Qin et al., 2023]. Therefore, the final core step is to transform the ordered demonstrations into a structured prompt $\mathcal{P}$, incorporating delimiters and instructions to optimize MM-ICL.

## 3 What Factors Affect Multi-modal In-Context Learning?

### 3.1 Exploration of MM-ICL Demonstration Retrieval

The efficacy of ICL heavily depends on the quality of the retrieved demonstrations $\mathcal{C}$, which provide essential prior knowledge for MM-ICL. As illustrated in Figure 2, the retrieval process encompasses

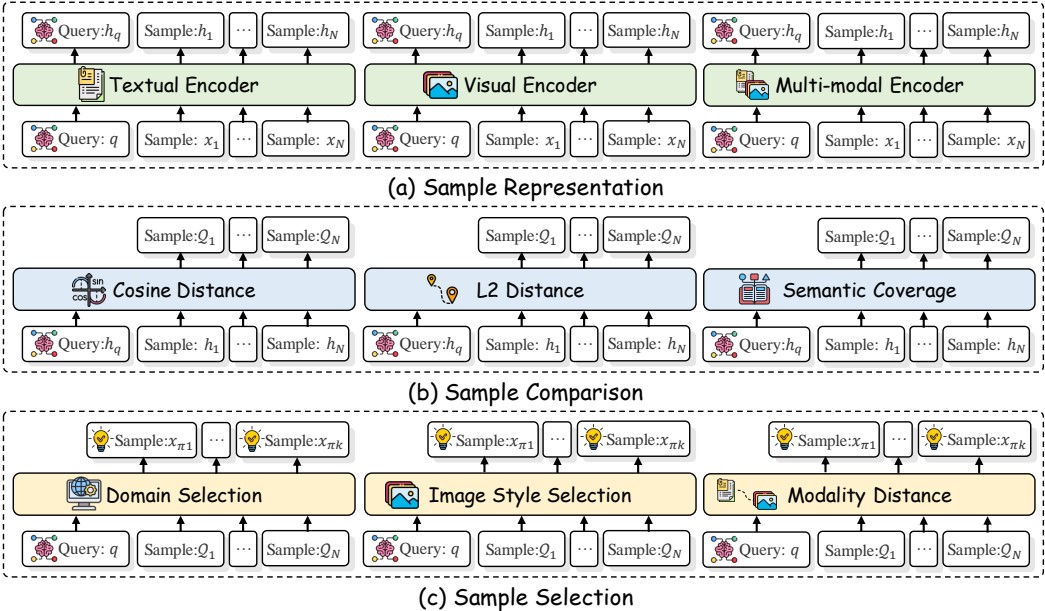

Figure 2: The demonstration retrieval process for MM-ICL.

three key steps: (1) Sample Representation, (2) Sample Comparison, and (3) Sample Selection. In this section, we conduct a systematic analysis of how various strategies for sample representation, comparison, and selection affect MM-ICL task performance.

**Sample Representation.** It involves defining an encoder ($\texttt{Encoder}(\cdot)$) to map each input sample $x_j \in \mathcal{V}$ and user query $q$ into a shared representation space:

$$h_j = \texttt{Encoder}(x_j). \tag{1}$$

Specifically, we evaluate various encoder architectures across modalities, focusing on the impact of visual encoder ($\texttt{Encoder}_{vis}$), text encoder ($\texttt{Encoder}_{txt}$), and multi-modal encoder ($\texttt{Encoder}_{multi}$) on model performance.

**Sample Comparison.** After deriving the representations, we employ a metric $\mathcal{M}$ to evaluate the quality $\mathcal{Q}_j$ of the sample $h_j$ in comparison to the query representation $h_q$ and the dataset samples $h_j$:

$$\mathcal{Q}_j = \mathcal{M}(h_q, h_j). \tag{2}$$

Specifically, we explore various comparison metrics, including cosine similarity $\mathcal{M}_{cos}$ [Liu et al., 2022a], L2 similarity $\mathcal{M}_{L2}$ [Liu et al., 2022a], and semantic diversity $\mathcal{M}_{div}$ [Li and Qiu, 2023a], to assess sample quality and understand the correlation with model performance.

**Sample Selection.** After quality assessments, we apply a selection criterion $\mathcal{S}$ to identify the $k$ most advantageous samples $x_{\pi_j}$ for inclusion in the demonstration set $\mathcal{C}$:

$$\mathcal{C} = \{x_{\pi_j} | x_{\pi_j} \in \mathcal{S}(q, \mathcal{Q}_j), j \le k\}. \tag{3}$$

Sample selection is guided by factors such as domain information [He et al., 2023], demonstration style [Agrawal et al., 2023], and token distance [Liu et al., 2022a]. Specifically, we systematically examine samples from both in-domain and out-of-domain collections. And we also assess the impact of image style on the selected demonstrations. Further, we investigate the token distance between modalities to understand its effects on sample selection for MM-ICL.

### 3.2 Exploration of MM-ICL Demonstration Ordering

Following Lu et al. [2022b] and Wu et al. [2023], the order of the demonstration set $\mathcal{C}$ significantly impacts MM-ICL performance. As shown in Figure 3, this section explores two key aspects:

**Intra-demonstration Ordering.** The sequence within a demonstration, especially modalities (e.g., text and image), is an important component that might affect the MM-ICL capabilities. Therefore,

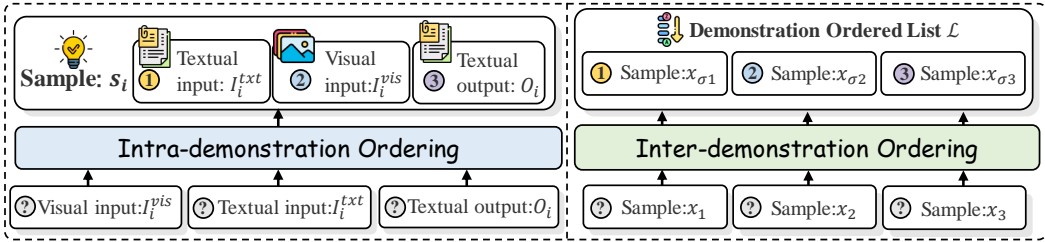

Figure 3: The demonstration ordering process for MM-ICL.

we introduce a intra-demonstration ordering permutation (IOP) to define this sequence:

$$\mathcal{L} = [\texttt{IOP}(x_{\pi_1}), \texttt{IOP}(x_{\pi_2}), \ldots, \texttt{IOP}(x_{\pi_k})]. \tag{4}$$

We conduct a systematic exploration of various IOP configurations, including text-image-text ($\texttt{IOP}^{\texttt{tvt}}$), text-text-image ($\texttt{IOP}^{\texttt{ttv}}$), and image-text-text ($\texttt{IOP}^{\texttt{vtt}}$). These order analyses aim to evaluate the impact of different modal sequences on the model's performance.

**Inter-demonstration Ordering.** The sequence in which demonstrations are organized within $\mathcal{C}$ also is the key component that might impact the performance of MM-ICL. Formally, we define a sample ordering permutation $\sigma_j$ to specify the arrangement:

$$\mathcal{L} = [x_{\sigma_1}, x_{\sigma_2}, \ldots, x_{\sigma_k} | x_{\sigma_j} \in \mathcal{C}], \tag{5}$$

where $x_{\sigma_j}$ represents the $j$-th demonstration in the ordered demonstration list.

### 3.3 Exploration of MM-ICL Prompt Construction

VLLMs are highly sensitive to input instructions [Kojima et al., 2022, Qin et al., 2023]. Inspired by this, to enhance task comprehension, we incorporate different instructions to explore the performance influence for MM-ICL. Formally, we construct instruction methods $\mathcal{I}(\cdot)$ that describe the task and position them within the prompt. The prompt construction process is:

$$\mathcal{P} = \mathcal{I}(\delta(x_{\sigma_1}), \delta(x_{\sigma_2}), \ldots, \delta(x_{\sigma_k})), \tag{6}$$

Specifically, as shown in Figure 4, we explore three instruction categories to bolster MM-ICL process:

- **Introductory Instruction ($\mathcal{I}_{intro}$)** refers to the initial guidance that offers an overview of the task prior to any demonstrations. As shown in Figure 4 (a), this instruction, denoted as $\mathcal{I}_{intro}$, is positioned at the start of the ordered demonstration sequence, $\mathcal{L}$.

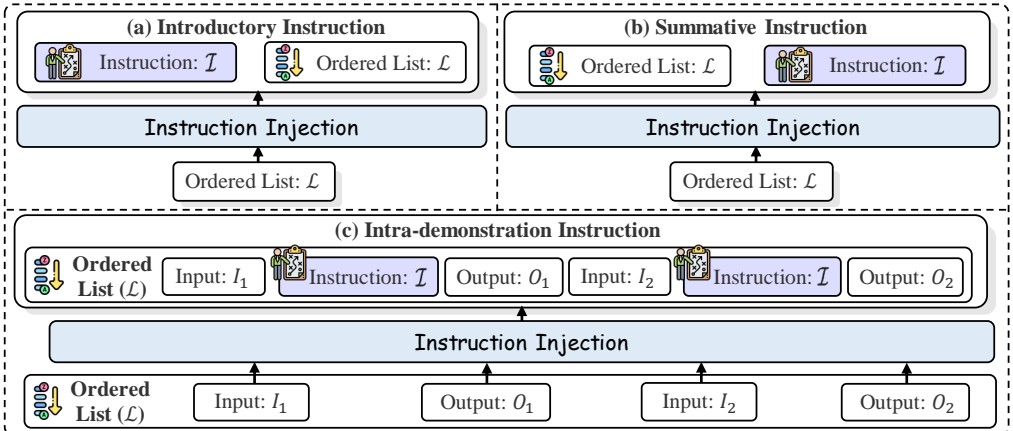

Figure 4: The process of instruction injection for MM-ICL prompt construction involves three key elements. The Introductory Instruction provides an overview instruction of the task before demonstrations. The Summative Instruction summarizes after the examples, guiding the model to apply the learned concepts to practical problems. The Intra-demonstration Instruction embeds task-specific guidance within each demonstration, enabling VLLMs to grasp task requirements during learning. Further details and additional prompts are provided in Appendix C.3.

- **Summative Instruction** ($\mathcal{I}_{sum}$) offers a summary after the examples, guiding the model to apply the learned concepts to real-world problems. As shown in Figure 4 (b), this instruction $\mathcal{I}$ is added at the end of the demonstration list $\mathcal{L}$.

- **Intra-demonstration Instruction** ($\mathcal{I}_{intra}$) embeds task instructions within each demonstration, helping VLLMs understand the task requirements during the learning process. As shown in Figure 4 (c), this instruction $\mathcal{I}$ is included within each demonstration $x_i$ in the list $\mathcal{L}$.

# 4 Experimental Setup

Following the setting of Li et al. [2023c], we systematically explore 4 tasks, including image-caption, visual question answering (VQA), image classification, and chain-of-thought reasoning, which come from M$^3$IT [Li et al., 2023c] and M$^3$CoT [Chen et al., 2024b] (as shown in Tables 2), providing a universal paradigm can help researchers conduct unified and fairer comparisons and studies within a unified framework. In order to evaluate the MM-ICL performance accurately, we use two indicators for each task. Following Zhang et al. [2019], Li et al. [2023b], and Zong et al. [2024], we use CIDER [Vedantam et al., 2015] and BertScore [Zhang et al., 2019] as image-caption metrics. Since M$^3$IT includes various VQA tasks with free-form answers, inspired by the success of free-form and precise answer hybrid evaluation in machine reading comprehension, following Rajpurkar et al. [2016], Zhang et al. [2019], we adapt Token-F1 [Rajpurkar et al., 2016] and BertScore as visual question answering (VQA) metrics (The correlation analysis of the indicators and accuracy as shown in Table 3). Following Li et al. [2023c,b], we use accuracy and F1 score as indicators of image classification. Following Lu et al. [2022a], Golovneva et al. [2022] and Qin et al. [2023], we use accuracy and reasoning alignment score [Golovneva et al., 2022] (RAS) as indicators of reasoning.

To ensure rigorous experimental control, we established a baseline using a multi-modal encoder for data representation and cosine similarity for sample comparison, limiting retrieval to within the same task. This baseline ranks samples based on similarity, with a delimiter and a 3-shot setting (see Appendix A for details). In addition, all open source models complete inference on 2 A100 80G. For all experiments, we select top-p from $\{0.95, 1\}$ and adjust the temperature parameter within $[0, 1]$. Among them, temperature is the main error variable in this work.

# 5 Empirical Analysis of Factors Affecting MM-ICL

## 5.1 Empirical Analysis of MM-ICL Demonstration Retrieval

### 5.1.1 Sample Representation

**Multi-modal alignment is the bottleneck for MM-ICL in both backbones and demonstrations.** To evaluate the impact of semantic representation in different modalities for MM-ICL, we assessed three distinct encoders: RoBERTa [Liu et al., 2019] as a textual encoder for `Textual Retriever`, CLIP-Vision Encoder [Radford et al., 2021] for `Visual Retriever`, and BridgeTower [Xu et al., 2023] as multi-modal encoder for `Multi-Modal Retriever`. As illustrated in Table 1, multi-modal retrieval consistently outperforms zero-shot, randomly selected, and single-modality methods, highlighting the advantages of multi-modal semantic learning for MM-ICL. What's more, as shown in Table 1, our results reveal that increasing model parameters from 8 billion to over 100 billion does not significantly enhance performance, suggesting that beyond parameter size, multi-modal context understanding and alignment are more crucial for MM-ICL than model scaling. Our analysis demonstrates that multi-modal alignment is the critical factor in both the backbone and demonstrations.

**Current multi-modal encoders still lack modeling of multi-modal logic.** Actually, multi-modal retrieval attains better performance in many scenarios like Image Caption and VQA. However, our experiments show that textual retrieval works well for classification and reasoning tasks. Based on the qualitative analysis, we observe that due to the semantic richness of the labels and rationales, textual retrieval can obtain more similar samples. However, the current multi-modal retrieval struggles with complex text semantics, often favoring image similarity. This aligns with recent work [Tong et al., 2023, 2024, Fei et al., 2024c], which is valuable for future exploration.

**Multi-modal context diminishes the necessity of careful demonstration selection.** As shown in Table 1, adding relevant demonstrations slightly improves performance, but the gains are less

| | Caption | | VQA | | Classification | | Reasoning | | AVG |
|---|---|---|---|---|---|---|---|---|---|
| | CIDER | BERTScore | Token F1 | BERTScore | Acc | F1 | Acc | RAS | |
| *OpenFlamingo (9B)* [Awadalla et al., 2023] | | | | | | | | | |
| Zero-shot | 1.84 | 81.18 | 2.78 | 76.17 | 15.17 | 3.63 | 16.53 | 85.13 | 35.30 |
| Few-shot (Random) | 8.23 | 56.63 | 12.63 | 67.37 | 13.11 | 5.11 | 21.35 | 86.53 | 33.87 |
| + Textual Retriever | 13.39 | 74.22 | **21.80** | 75.74 | 13.67 | **12.04** | **25.63** | 87.71 | 40.52 |
| + Visual Retriever | 6.33 | 53.88 | 12.82 | 68.76 | 13.67 | 10.87 | 23.10 | 87.36 | 34.60 |
| + Multi-Modal Retriever | **13.47** | **85.01** | 7.85 | **79.05** | 19.66 | 10.10 | 24.96 | **88.11** | **41.03** |
| *Otter (9B)* [Li et al., 2023b] | | | | | | | | | |
| Zero-shot | 2.86 | 86.42 | 20.90 | 87.95 | 24.34 | 10.85 | 34.06 | 82.67 | 43.76 |
| Few-shot (Random) | 3.50 | 86.62 | 20.95 | 87.76 | 25.66 | 10.28 | 34.23 | 83.67 | 44.08 |
| + Textual Retriever | 3.89 | 86.62 | 20.40 | 87.89 | 25.28 | 8.47 | 32.88 | 81.93 | 43.42 |
| + Visual Retriever | 3.50 | 86.51 | 18.57 | 87.58 | 26.78 | 12.44 | 32.21 | 84.17 | 43.97 |
| + Multi-Modal Retriever | 3.77 | 86.57 | 18.80 | 87.56 | 28.65 | 11.83 | 35.92 | 83.74 | 44.60 |
| *Qwen-VL (10B)* [Bai et al., 2023] | | | | | | | | | |
| Zero-shot | 13.57 | 87.65 | 24.96 | 85.09 | 50.19 | 54.28 | **48.40** | 90.87 | 56.87 |
| Few-shot (Random) | 28.52 | 88.47 | 28.43 | 86.11 | 52.43 | 53.50 | 43.34 | 90.19 | 58.87 |
| + Textual Retriever | 21.58 | 88.07 | 26.99 | 85.62 | 49.44 | 53.04 | 46.04 | 90.72 | 57.69 |
| + Visual Retriever | 30.81 | 88.56 | 28.79 | 86.23 | 59.74 | **54.43** | 46.88 | 91.30 | 60.84 |
| + Multi-Modal Retriever | **41.51** | **89.03** | **30.20** | **86.78** | 59.36 | 53.17 | 46.21 | **91.49** | **62.22** |
| *GPT4V (>100B)* [OpenAI: et al., 2023] | | | | | | | | | |
| Zero-shot | 5.15 | 85.43 | 20.01 | 84.77 | 61.42 | 59.07 | 54.64 | 92.46 | 57.87 |
| Few-shot (Random) | 6.37 | 85.95 | 24.43 | 85.42 | 60.11 | 60.81 | 54.30 | 92.54 | 58.74 |
| + Textual Retriever | 9.48 | 86.02 | 31.81 | **87.02** | 62.55 | 51.40 | 55.99 | 92.26 | 59.57 |
| + Visual Retriever | 9.36 | 86.26 | 32.47 | 86.96 | **63.30** | 57.79 | 59.87 | **93.19** | 61.15 |
| + Multi-Modal Retriever | **16.55** | **86.77** | **32.92** | 86.87 | 62.55 | **59.97** | **60.88** | 93.10 | **62.45** |
| *IDEFICS2 (8B)* [Laurençon et al., 2024b] | | | | | | | | | |
| Zero-shot | 32.80 | 88.59 | 26.88 | 86.99 | **66.85** | 57.84 | **54.97** | 89.01 | 62.99 |
| Few-shot (Random) | 39.68 | 88.88 | 30.82 | 87.59 | 61.61 | 53.39 | 51.94 | 89.52 | 62.93 |
| + Textual Retriever | 35.95 | 88.45 | 31.66 | 87.58 | 61.99 | **67.13** | 45.03 | 88.98 | 63.34 |
| + Visual Retriever | 46.61 | 89.55 | 32.05 | 87.92 | 64.04 | 62.51 | 51.26 | **89.83** | 65.47 |
| + Multi-Modal Retriever | **52.55** | **89.66** | **33.65** | **88.17** | 65.54 | 63.86 | 51.43 | 89.57 | **66.80** |
| *Gemini-Pro (>100B)* [Google, 2023] | | | | | | | | | |
| Zero-shot | 14.05 | 87.07 | 26.93 | 85.78 | 68.20 | 66.10 | 55.14 | 90.72 | 61.75 |
| Few-shot (Random) | 21.21 | 88.13 | 32.99 | 86.81 | 63.67 | 69.75 | 55.65 | 91.82 | 63.75 |
| + Textual Retriever | 15.79 | 87.75 | 34.96 | 87.18 | **69.10** | **72.31** | 53.29 | 91.57 | 63.99 |
| + Visual Retriever | 21.35 | 87.98 | 44.74 | 89.33 | 65.73 | 64.18 | 53.29 | 91.92 | 64.81 |
| + Multi-Modal Retriever | **35.64** | **88.67** | **45.47** | **89.61** | 65.17 | 70.51 | **58.01** | **92.17** | **68.16** |

Table 1: Performance comparison of retrievers utilizing different modal representations, where `Few-shot (Random)` refers to MM-ICL methods in which the demonstrations are randomly selected from the development set.

significant compared to text-only ICL scenarios. Specifically, retrieved demonstrations yield an average performance boost of 3.84%, compared to random demonstrations. In contrast, text-only scenarios show performance increases of over 10% with carefully selected samples [Shi et al., 2023]. Furthermore, the model remains unaffected by irrelevant samples, and the performance of almost all models is higher than zero-shot. This indicates that multi-modal context significantly reduces the need for careful demonstration selection, unlike in text-only scenarios.

**VLLMs learn semantic representations instead of token pattern representations for MM-ICL.** As depicted by Agrawal et al. [2023], textual ICL primarily learns token patterns (e.g., similar output formats, reasoning paths) among demonstration outputs. To investigate whether VLLMs rely on repetitive token patterns, we utilize the average BLEU score across demonstration outputs as a representation of token repetition. Figure 5 shows that only the image captioning task exhibits a positive correlation. In contrast, other tasks show a decline as BLEU scores exceed 30%. This underscores that MM-ICL primarily learns semantic rather than token pattern representations for effective performance.

### 5.1.2 Sample Comparison

To further analyze the influencing factors of MM-ICL in sample retrieval, this study employs similarity and diversity metrics, which help assess how MM-ICL processes sample similarities and differences, enhancing our understanding of its mechanisms. See Appendix B for more details and results.

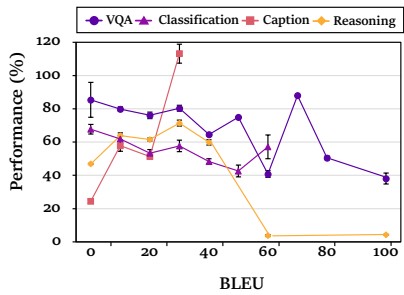

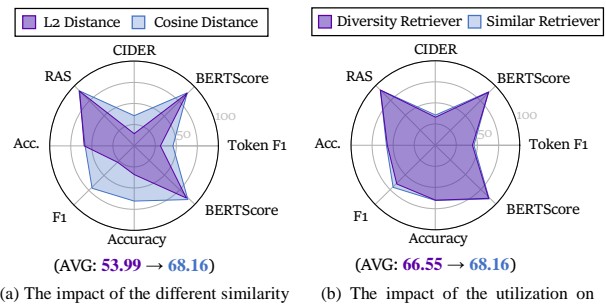

Figure 5: The impact of token pattern representation in Gemini-Pro.

(a) The impact of the different similarity metrics.

(b) The impact of the utilization on diversity metrics.

Figure 6: The impact of different sample comparison methodologies in Gemini-Pro.

**Cosine similarity matters for sample comparison.** Following Liu et al. [2022b], we compare two representative similarity metrics, cosine similarity and L2 similarity. As shown in Figure 6 (a), cosine similarity, which measures the directional semantic alignment, emerges as the superior metric in MM-ICL than L2 similarity. Supported by Deza et al. [2009] and Steck et al. [2024], it indicates that MM-ICL prioritizes semantic directional consistency over complete semantic alignment.

**Diversity does not show significant influence for sample comparison.** He et al. [2023], Li and Qiu [2023b] have shown that demonstrations with better diversity can effectively improve textual ICL. To explore whether it exists in MM-ICL, following Li and Qiu [2023b], we ultilize the "diversity retriever", which selects the top-10 samples and further chooses the best 3 samples based on semantic diversity to obtain a more diverse MM-ICL. As demonstrated in Figure 6 (b), although diversity significantly enhances performance in text-based ICL, our experiments show limited improvement in MM-ICL tasks. This suggests that diversity may not directly correlate with better MM-ICL.

### 5.1.3 Sample Selection

**Domain interval matters for sample selection.** Prior research highlights the critical role of domain relevance in enhancing ICL performance. Inspired by this, we employ the multi-modal retriever to select samples from both in-domain and out-of-domain pools. Figure 7 (a) shows a nearly 4% performance drop when out-of-domain demonstrations are included, underscoring the necessity of in-domain demonstrations for optimal MM-ICL.

**Visual style is not a crucial factor in sample selection.** Although stylistic similarity in text samples is known to bolster ICL, its effect on the visual modality remains ambiguous. Utilizing CLIP for image classification, we investigate the impact of stylistic coherence in multi-modal samples on MM-ICL performance. As depicted in Figure 7 (b), significant enhancements are observed solely in the VQA task, while captioning and classification show minimal effects and reasoning tasks decline. This indicates that diverse visual styles are not crucial in general MM-ICL.

**Token distances between modalities need to be considered for different tasks to improve sample selection.** For textual ICL, excessive token distance between samples can impede performance [Liu

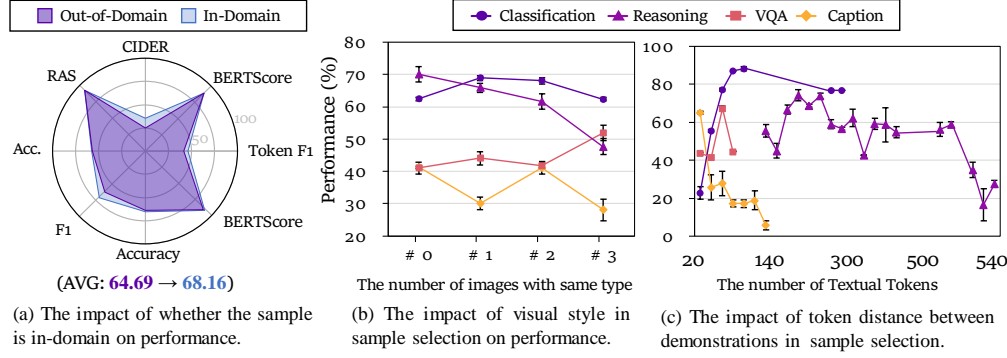

(a) The impact of whether the sample is in-domain on performance.

(b) The impact of visual style in sample selection on performance.

(c) The impact of token distance between demonstrations in sample selection.

Figure 7: The impact of sample selection on average score performance in Gemini-Pro.

et al., 2022a]. We extend this inquiry to MM-ICL, analyzing how token distance across modalities influences results. Specifically, during the sample selection process, we considered the impact of the average token distance between two images on the model within the entire prompt of MM-ICL. As illustrated in Figure 7 (c), the effect of token distance varies by task, typically showing an initial performance increase followed by a decline as distance grows, particularly in non-captioning tasks. This highlights the task-dependent nature of optimal token distance in MM-ICL.

## 5.2 Empirical Analysis of MM-ICL Demonstration Ordering

**Intra-demonstration ordering significantly impacts performance.** Within the demonstration, organizing the ordering, especially the relationship between modalities is a crucial topic. We investigate this by arranging inputs and outputs across modalities using three methods: *text input→text output→image input* (Text-Image), *text input→image input→text output* (Text-Image-Text), and *image input→text input→text output* (Image-Text). As shown in Figure 8 (a), positioning the image at the start significantly enhances model performance. This suggests that presenting visual information first improves multi-modal comprehension, thereby boosting its learning abilities.

**Inter-demonstration ordering demonstrates minimal impacts.** Following Lu et al. [2022c], we investigate how the order of demonstration presentation influences model efficacy. We explore various strategies: random rearrangement, a "similar-last" approach where samples similar to the query are shown last, and a "similar-first" approach where similar samples are presented first. Figure 8 (b) illustrates that inter-demonstration ordering has a negligible impact on MM-ICL performance. This suggests the order-robustness, with the presentation sequence having minimal effect.

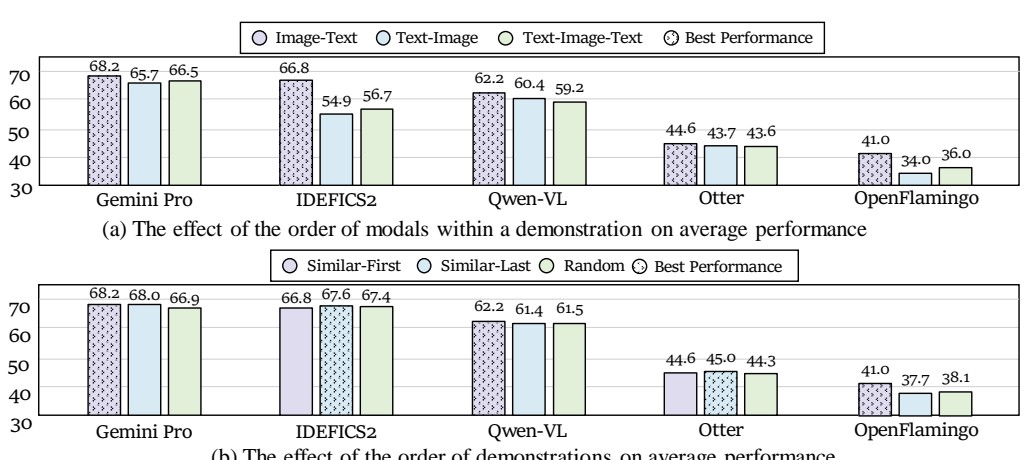

(a) The effect of the order of modals within a demonstration on average performance

(b) The effect of the order of demonstrations on average performance

Figure 8: The impact of demonstration ordering on performance.

## 5.3 Empirical Analysis of MM-ICL Prompt Construction

**Introductory Instruction is consistently effective for better MM-ICL.** To investigate the impact of inserting task-related instructions within prompts, we conduct the following experiment on three categories of instruction: *Introductory Instruction*, *Summative Instruction*, and *Intra-demonstration Instruction*. As depicted in Figure 9, our analysis indicates that introductory instructions stably

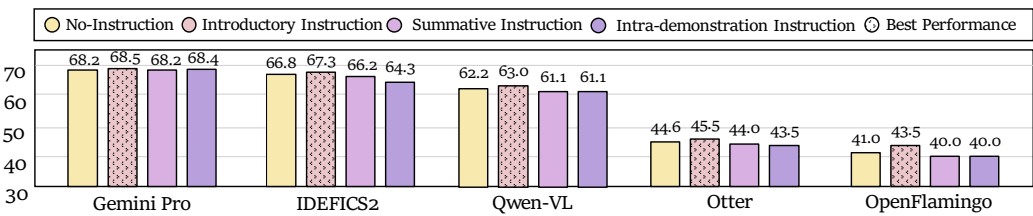

Figure 9: The impact of injecting instruction into demonstrations on model average score performance.

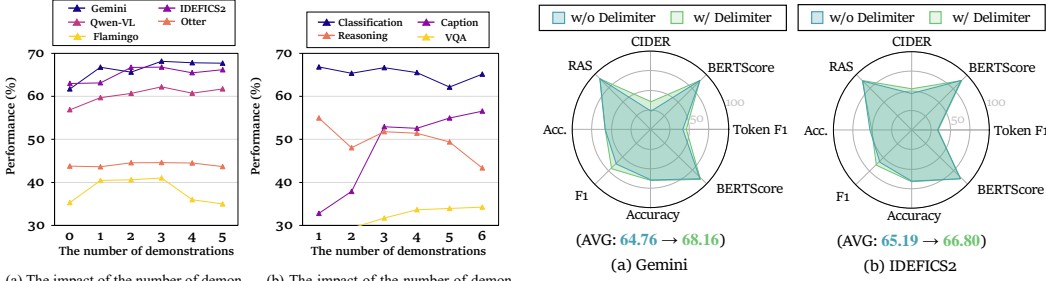

(a) The impact of the number of demonstrations on average score performance.

(b) The impact of the number of demonstrations on different task performance.

Figure 10: The impact of the number of demonstrations on performance.

Figure 11: The impact of inserting delimiter into the input and output of demonstration on model performance. See Figure 16 for more results.

enhance model performance. In contrast, other instructions generally decrease performance. This finding suggests that introductory instructions facilitate targeted contextual learning and more effective semantic comprehension in demonstrations. We show more prompts and details in Appendix C.3.

**MM-ICL is affected by the number of demonstrations depending on the task.** Contrary to traditional text-based ICL, where performance improves with more samples, our findings in Figure 10 (a) suggest that MM-ICL does not experience significant gains from more demonstrations. To further understand the reason behind, we analysis the performance on different tasks. As shown in Figure 10 (b), increasing the number of demonstrations enhances performance in caption and VQA tasks, a trend also reported in prior studies [Alayrac et al., 2022, Laurençon et al., 2024a, Shukor et al., 2024]. However, performance declines when demonstrations exceed three across all tested VLLMs. In more complex reasoning tasks, such as multi-step multi-modal chain-of-thought reasoning, additional demonstrations do not yield effective improvements, aligning with the findings of Chen et al. [2024b], and Fei et al. [2024a].

Moreover, we attribute it to the following reasons for this limitation: (1) *Cognitive Overload:* For complex tasks, understanding numerous demonstrations can overwhelm the model, impeding its ability to process and integrate information effectively [Chen et al., 2024a]. (1) *Complexity of Reasoning Tasks:* In reasoning tasks, the performance improvement from more demonstrations is often less pronounced than when using diverse retrievers. This suggests that reasoning tasks require sophisticated integration of information, where quality outweighs quantity. See Appendix C.1 for more detailed description.

**The importance of delimiter lessens by text-image interleaved demonstrations.** Previous research suggests that specific delimiters for input and output data can demonstrably influence textual ICL capabilities [Min et al., 2022]. Therefore, we utilize ablation experiments to omit these delimiters to examine their necessity (see Appendix C.2 for details). As shown in Figure 11, the resulting minor performance decline suggests that while these delimiters are less critical in MM-ICL, the modality switch inherent to MM-ICL may serve as an implicit delimiter, compensating for the absence of explicit delimiters.

## 6  Related Work

Recent advancements in vision large language models (VLLMs) have achieved great success in various vision-language tasks [Yin et al., 2023, Wu et al., 2024a,b, Wang et al., 2024, Fei et al., 2024b]. Initially, VLLMs lack Multi-modal In-context Learning (MM-ICL) capabilities. To address this, researchers explore incorporating MM-ICL directly into the training phase. This involves constructing training samples with multi-modal interleaved data by manual and general templates, which unlock the MM-ICL capability [Alayrac et al., 2022, Awadalla et al., 2023]. Building on this, Li et al. [2023b], Doveh et al. [2024] and Zhao et al. [2024] extend the MM-ICL to construct a series of task-specific templates, which improves generalization for MM-ICL. Further, Li et al. [2023a] introduce OtterHD and adapt the former process for high-definition images. The potential of MM-ICL is further explored in scene text recognition, image generation, and game instructions [Zhao et al., 2023b, Sun et al., 2023, Jin et al., 2024].

Recognizing the effectiveness of MM-ICL, researches shift towards prompt optimization. These methods focus on directly optimizing multi-modal prompts to understand the task and generate the expected output, without parameter adjustments [Gong et al., 2023, Tsimpoukelli et al., 2021, Li et al., 2023b]. This approach has significantly improved performance in visual reasoning tasks [Yang et al., 2022, Zheng et al., 2023]. Another approach involves textualizing visual information to enable VLLMs to leverage their background knowledge through in-context learning, further enhancing visual reasoning [Yang et al., 2023, Lu et al., 2024, Gupta and Kembhavi, 2023, Shen et al., 2024]. In addition, in order to better explore the MM-ICL, Zong et al. [2024] and Shukor et al. [2024] also provide a dataset to test the MM-ICL capabilities of the multi-modal classification. Furthermore, Shukor et al. [2024] take the first step to conduct an instruction modification exploration for MM-ICL.

Meanwhile, Baldassini et al. [2024], Chen et al. [2024c] pioneer the first naive multi-modal retrieval exploration to enhance MM-ICL. Different from the existing work, our study mainly focuses on a **systematic exploration** of the effectiveness of key factors influencing the effectiveness of MM-ICL in a **unified perspective**. To this end, we conduct a detailed analysis and exploration on 6 VLLMs and 20 factors across 4 tasks, aiming to provide systematic and practical guidance for future research.

# 7 Discussion

**Broader Impacts.** Our work is the first to systematically explore the factors influencing MM-ICL. We aim to enhance the understanding of MM-ICL mechanisms and guide future developments in this field. Additionally, our findings could foster a more comprehensive comprehension of MM-ICL within the community. For social impact, this research may influence the creation of more effective multi-modal large language models and relevant applications.

**Limitations & Future Work.** Due to time and cost constraints, this work is limited to the exploration of image and text modalities. In future research, we can extend our exploration to video modal ICL and multi-lingual MM-ICL scenarios. Another limitation of this work involves the insufficient consideration of certain image instructions, such as grounding or the inclusion of additional arrows. These aspects often require more complex human input and are not adequately supported by most current models.

# 8 Conclusion

This study is the first to systematically explore MM-ICL by identifying key performance determinants. Our experiments with 6 models and 20 factors across 4 tasks show that multi-modal retrieval significantly outperforms single-modal approaches and the intra-demonstration ordering critically influences learning efficacy. Additionally, incorporating task-specific instructions into prompts enhances model performance. We hope these findings will refine our understanding of MM-ICL mechanisms and guide more effective developments and future research in this evolving field.

## Acknowledgments

This work was supported by the National Natural Science Foundation of China (NSFC) via grant 62306342, 62236004, 62441603 and 62476073. This work was also sponsored by the Excellent Young Scientists Fund in Hunan Province (2024JJ4070) and the Science and Technology Innovation Program of Hunan Province under Grant 2024RC3024. We are grateful for resources from the High Performance Computing Center of Central South University, and the CCF-Zhipu.AI Large Model Innovation Fund (NO.CCF-Zhipu202406). Libo Qin is the corresponding author.

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

# Appendix

## A  The Implement Details for Standard Baseline

To ensure rigorous control of experimental variables, we establish a standard baseline for our study. This baseline utilizes a multi-modal encoder for data representation and cosine similarity for sample comparison, with retrieval restricted to the same task. The following sections provide detailed insights into the implementation of this baseline.

### A.1  Demonstration Retrieval Implementation for Baseline

**Text-modal Encoder for Sample Representation**    We employ RoBERTa [Liu et al., 2019] as a multi-modal encoder to represent the data in a unified embedding space. It is worth noting that since the Image Caption task only contains a general task instruction, we will splice the field tags contained in the meta data in M3IT before the insrtuction for comprehensive text representation.

**Multi-modal Encoder for Sample Representation**    We employ BridgeTower [Xu et al., 2023] as a multi-modal encoder to represent the data in a unified embedding space. This encoder integrates both visual and textual information, prioritizing single modality to capture the rich semantic content present in images.

**Cosine Similarity for Sample Comparison**    To compare samples effectively, we use cosine similarity, a metric that measures the cosine of the angle between two non-zero vectors in a multi-dimensional space. This choice is motivated by its effectiveness in capturing the similarity between high-dimensional vectors, which are typical outputs of our multi-modal encoder. Specifically, we compute the cosine similarity between the query and each candidate sample, which is given by:

$$\text{cosine}(h_q, h_i) = \frac{h_q \cdot h_i}{\|h_q\|\|h_i\|}$$

where $h_q$ and $h_i$ are the embedding vectors of the query $q$ and candidate sample $x_i$, respectively.

**In-domain and Top-k Retrieval for Sample Selection**    To ensure the relevance and accuracy of the retrieval process, for sample selection, we first confine retrieval to the same task and domain. This means that comparisons and rankings are conducted exclusively among samples within the same task and domain category, ensuring the contextual appropriateness of the retrieved results.

| Dataset | Category |
|---|---|
| COCO Caption [Chen et al., 2015] | IC |
| TextCaps [Sidorov et al., 2020] | IC |
| Paragraph Captioning [Krause et al., 2017] | IC |
| COCO Text [Veit et al., 2016] | CLS |
| ImageNet Image Classification [Russakovsky et al., 2015] | CLS |
| IQA [Duanmu et al., 2021] | CLS |
| COCO-ITM [Chen et al., 2015] | CLS |
| e-SNLI-VE [Kayser et al., 2021] | CLS |
| Mocheg [Yao et al., 2023] | CLS |
| VQA-v2 [Goyal et al., 2017] | VQA |
| DocVQA [Mathew et al., 2021] | VQA |
| OCR-VQA [Mishra et al., 2019] | VQA |
| ST-VQA [Biten et al., 2019] | VQA |
| Text-VQA [Singh et al., 2019] | VQA |
| GQA [Hudson and Manning, 2019] | VQA |
| OKVQA [Marino et al., 2019] | VQA |
| A-OKVQA [Schwenk et al., 2022] | VQA |
| ScienceQA [Lu et al., 2022a] | R |
| M$^3$CoT [Chen et al., 2024b] | R |

Table 2: Dataset in M$^3$IT and M$^3$CoT, where IC: Image Captioning, CLS: Classification, VQA: Visual Question Answering, R: Chain-of-Thought Reasoning (with NL rationale). Due to the cost, for each task, we evenly sampled 500 items according to the sub-dataset.

In addition, for sample selection, samples are ranked according to their cosine similarity scores. Higher similarity scores indicate a closer alignment with the query sample, enabling the efficient identification of the most relevant samples. This ranking process involves two main steps: (1) Sorting: Candidate samples are sorted in descending order based on their cosine similarity scores relative to the query. (2) Selection: Subsequently, the top-k ranked samples are selected based on their relevance as determined by the similarity scores.

## A.2 Demonstration Ordering Implementation for Baseline

By default, we utilize the methodology for ordering demonstrations within our baseline model. By default, we adopt a text-after-image (`Text-Image`) approach for intra-demonstration sorting. This means that, within a single demonstration, textual information is positioned after the corresponding image. This ordering is chosen based on preliminary findings suggesting that such a sequence aids in better contextual understanding and retention of the demonstrated information.

Furthermore, for the ordering of inter-demonstration sequences, we employ a similarity-based method. This method ranks demonstrations according to their similarity to the query, with more similar demonstrations placed higher in the order. The similarity is determined using a metric that assesses the alignment of key features between the query and the demonstrations. This approach ensures that the most relevant and contextually aligned demonstrations are prioritized, potentially enhancing the model's performance and the user's comprehension.

## A.3 Prompt Construction Implementation for Baseline

To ensure consistency and comparability in our baseline, we introduce both a delimiter and a 3-shot setting (following Wei et al. [2022b], Qin et al. [2023]). The delimiter serves to clearly demarcate different segments of the input data, preventing any potential confusion or overlap between distinct portions of the input. This clear separation is crucial for the model to accurately process and understand the structure of the data it receives.

The 3-shot setting, on the other hand, involves providing three examples for each task within the prompt. This approach is designed to stabilize the learning process by presenting the model with sufficient contextual information. By offering three examples, we strike a balance between providing enough context to guide the model's understanding and avoiding the cognitive overload that might occur with too many examples. This setting not only enhances the model's performance but also ensures a more robust and reliable learning process.

## A.4 Baseline Prompt

In the context of using Vision-and-Language Large Models (VLLMs), it is essential to carefully structure the input prompts to ensure accurate processing. The prompt format typically used is illustrated below:

---

**[REQUEST]** % Shot 1

  *<Visual Input $\mathcal{I}_1^{vis}$>*   *<Textual Input $\mathcal{I}_1^{txt}$>*

**[RESPONSE]**

  *<Textual Output $\mathcal{I}_1^{vis}$>*

**[REQUEST]** % Shot 2

  *<Visual Input $\mathcal{I}_2^{vis}$>*   *<Textual Input $\mathcal{I}_2^{txt}$>*

**[RESPONSE]**

  *<Textual Output $\mathcal{I}_2^{vis}$>*

**[REQUEST]** % Shot 3

  *<Visual Input $\mathcal{I}_3^{vis}$>*   *<Textual Input $\mathcal{I}_3^{txt}$>*

---

> **[RESPONSE]**
>
> *<Textual Output $\mathcal{I}_3^{vis}$>*
>
> **[REQUEST]** % User Query
>
> *<Visual Input $\mathcal{I}_q^{vis}$>*   *<Textual Input $\mathcal{I}_q^{txt}$>*

where any gray text following the percent sign (%) is treated as a comment. These comments are not processed as part of the primary input but serve to provide additional context or instructions within the coding environment. This convention helps in maintaining the clarity and functionality of the given prompting.

In conclusion, the standard baseline established here integrates a multi-modal encoder, cosine similarity, and task-specific retrieval with a focus on visual modalities. It ranks samples based on similarity and employs a delimiter with a 3-shot setting to ensure robust and consistent performance across different tasks.

## B  The Implement Details for Sample Comparison

### B.1  Metric Calculation

**Cosine Similarity ($\mathcal{M}_{cos}$)**   Compute the cosine similarity between $h_q$ and $h_j$ using the formula:

$$\mathcal{M}_{cos}(h_q, h_j) = \frac{h_q \cdot h_j}{\|h_q\|\|h_j\|} \tag{7}$$

**L2 Similarity ($\mathcal{M}_{L2}$)**   Calculate the L2 similarity by computing the negative Euclidean distance between $h_q$ and $h_j$:

$$\mathcal{M}_{L2}(h_q, h_j) = -\|h_q - h_j\|_2 \tag{8}$$

Since Euclidean distance measures dissimilarity, we use the negative value to represent similarity, where a higher value indicates greater similarity.

**Semantic Diversity ($\mathcal{M}_{div}$)**   Semantic diversity is assessed by evaluating the differences in the distributional properties of $h_q$ and $h_j$. This assessment involves analyzing the variance in how these properties are distributed across different samples. To determine the presence of semantic diversity within Multi-Modal In-Context Learning (MM-ICL), we adopt the methodology proposed by Li and Qiu [2023b]. Specifically, we employ the "diversity retriever," designed to enhance the diversity of the selected samples. The diversity retriever operates by first selecting the top 10 samples based on a preliminary measure of relevance. From these top 10 samples, it then identifies the 3 samples that exhibit the highest semantic diversity. This two-step process ensures that the final selection of samples for MM-ICL is not only relevant but also diverse in terms of their semantic content.

### B.2  Comparison and Analysis

Comparing the results obtained using different metrics ($\mathcal{M}_{cos}$, $\mathcal{M}_{L2}$, $\mathcal{M}_{div}$) provides a comprehensive understanding of their effectiveness and suitability for specific applications. It is essential to analyze the trade-offs associated with each metric and interpret the results to draw meaningful conclusions about sample quality and relevance.

As shown in Figure 12, cosine similarity, which measures directional semantic alignment, emerges as the superior metric in MM-ICL compared to L2 similarity. This observation is supported by the findings of Deza et al. [2009] and Steck et al. [2024], who highlight that MM-ICL prioritizes semantic directional consistency over complete semantic alignment. Cosine similarity's ability to capture the nuances of directional alignment allows for more precise interpretations of semantic relationships within the data, making it particularly effective for MM-ICL tasks.

In contrast, Figure 13 illustrates that while diversity, as measured by $\mathcal{M}_{div}$, enhances performance in text-based in-context learning, our experiments reveal limited improvement in MM-ICL tasks. This finding suggests that diversity may not directly correlate with better performance in MM-ICL. The limited impact of diversity on MM-ICL performance could be attributed to the specific nature of

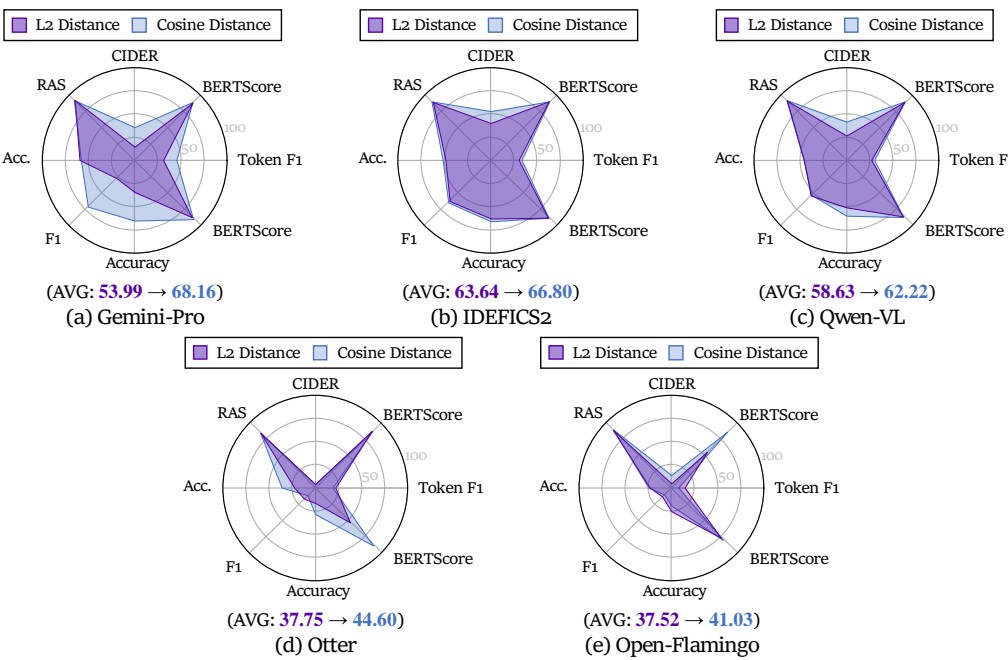

Figure 12: The impact of the different similarity metrics.

multi-modal data, where the interplay between different modalities requires a more nuanced approach than simply maximizing diversity.

Further analysis of these metrics reveals the inherent trade-offs between them. For instance, while cosine similarity offers advantages in maintaining semantic directional consistency, it may not capture the full extent of semantic similarity that L2 similarity can provide. On the other hand, L2 similarity, though comprehensive in measuring complete alignment, might lack the precision needed for tasks that rely heavily on directional semantic cues. Similarly, while diversity is beneficial in certain contexts, its role in MM-ICL needs to be reconsidered, potentially focusing on optimizing other aspects of sample quality.

In summary, the evaluation of $\mathcal{M}_{cos}$, $\mathcal{M}_{L2}$, and $\mathcal{M}_{div}$ underscores the importance of selecting appropriate metrics based on the specific requirements of the task. Understanding the trade-offs and context-specific effectiveness of these metrics is crucial for optimizing performance in multi-modal in-context learning applications.

## C   Exploration of MM-ICL Prompt Construction

### C.1   The Implement Details for Demonstration Sampling

To examine the effect of demonstration sample quantity on model performance, as shown in Figure 14, we select a subset of $k'$ demonstrations from the demonstration list $\mathcal{L}_{k'}$ to the prompt, where $k'$ is the number of retrieved demonstrations. Formally, the prompt construction process is defined as:

$$\mathcal{P} = \mathcal{I}(\delta(x_{\sigma_1^j}), \delta(x_{\sigma_2^j}), \ldots, \delta(x_{\sigma_{k'}^j})) \tag{9}$$

We systematically evaluate the influence of varying $k'$ on MM-ICL performance.

### C.2   The Implement Details for Delimiter Injection

To distinctly separate inputs and outputs within demonstrations $x_i$, as shown in Figure 15, we leverage special delimiter markers. Delimiters like [Request] and [Response] are strategically placed before the inputs and outputs, respectively. Formally, delimiter injection function $\delta$ maps inputs and outputs to the prompting sequences:

$$\delta(x_{\sigma_i}) = [\text{Request}] \oplus I_i \oplus [\text{Response}] \oplus O_i, \tag{10}$$

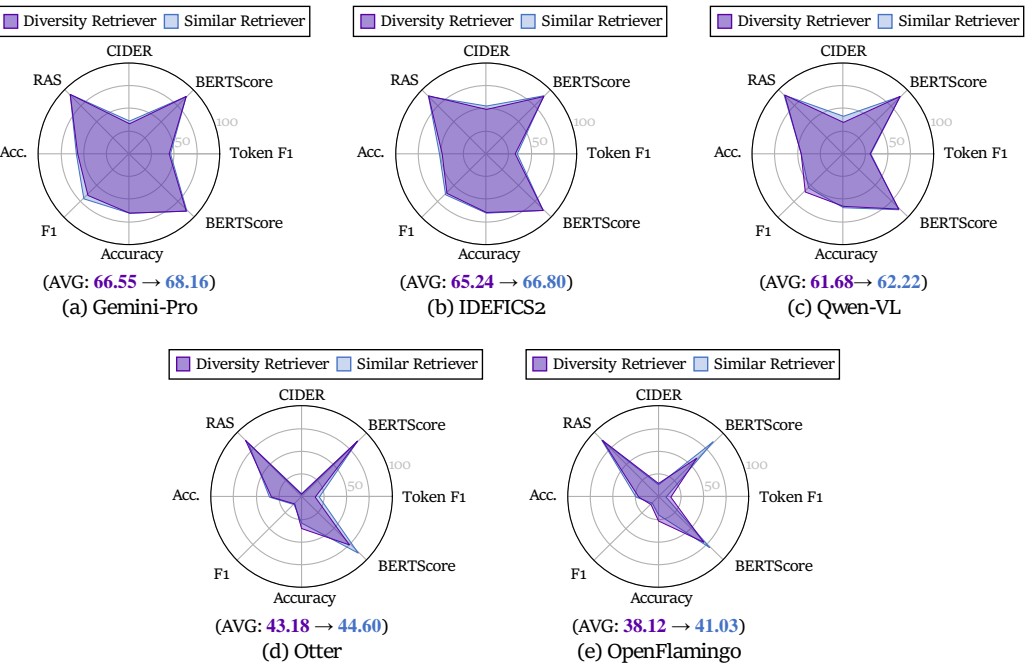

Figure 13: The impact of the utilization on diversity metrics.

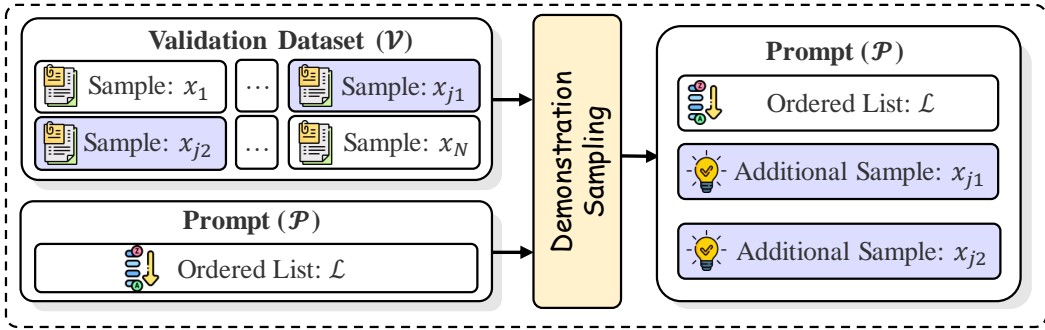

Figure 14: The demonstration sampling process for MM-ICL prompt construction.

where $I_i$ and $O_i$ denotes the input and output for the sample $x_i$, respectively. In addition, $\oplus$ represents string concatenation operation.

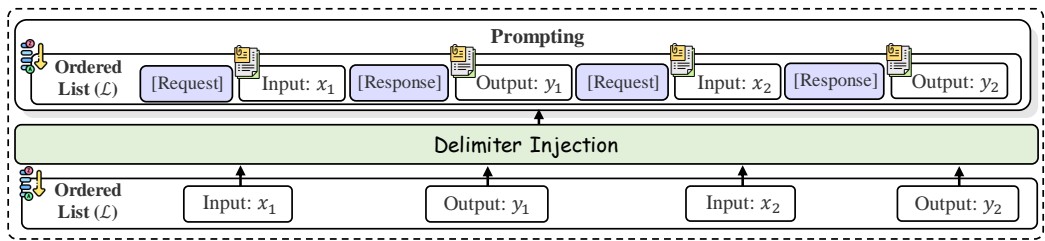

Figure 15: The delimiter injection process for MM-ICL prompt construction.

## C.3 The Implement Details for Instruction Injection

Visual Language Models (VLLMs) are known to be highly sensitive to input instructions, as demonstrated by Kojima et al. [2022] and Qin et al. [2023]. Inspired by this observation, we aim to enhance

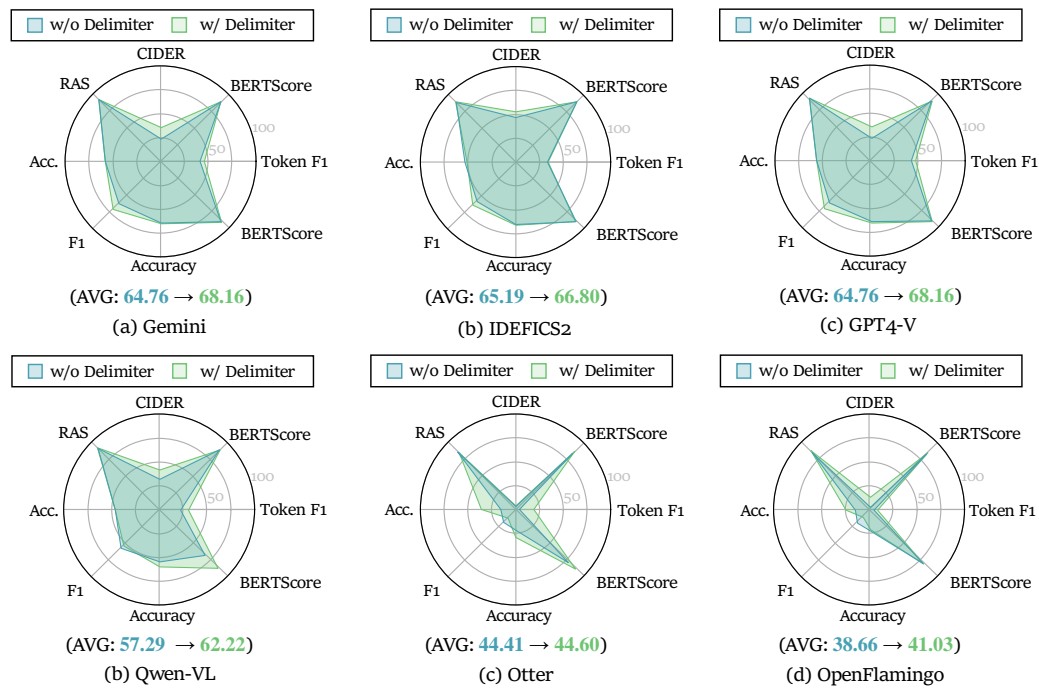

Figure 16: The impact of inserting delimiter into the input and output of demonstration on model performance.

task comprehension in Multi-Modal In-Context Learning (MM-ICL) by incorporating various instructions to explore their influence on performance. Formally, we develop instruction methods, denoted as $\mathcal{I}(\cdot)$, which describe the task and are integrated into the prompt construction process. The prompt $\mathcal{P}$ is constructed as follows:

$$\mathcal{P} = \mathcal{I}(\delta(x_{\sigma_1}), \delta(x_{\sigma_2}), \ldots, \delta(x_{\sigma_k})), \tag{11}$$

where $\delta(x_{\sigma_i})$ represents the transformation of the $i$-th demonstration example.

Specifically, we have designed distinct instructions tailored to different types of tasks, ensuring clarity and appropriateness for each unique context. For image captioning tasks, the prompt is:

> Please provide a caption for the image following the structure of the provided example.

In this context, the objective is to generate descriptive captions that accurately reflect the content and context of the image. For Visual Question Answering (VQA) tasks, our prompt is:

> Examine the image and answer the question by closely following the structure shown in the example provided.

The VQA tasks require the model to analyze visual content and respond to specific queries. By following the example, users can produce answers that are precise and directly related to the visual stimuli. For image classification tasks, the prompt is:

> Carefully review the image and categorize it based on the options provided in **[REQUEST]**, following the classification format illustrated in the example.

Image classification involves categorizing images into predefined classes based on visual content. The provided example demonstrates the expected classification format. For chain-of-thought reasoning tasks, the prompt is:

> Carefully review the given image and the associated text. Utilize the reasoning format illustrated in the provided examples, breaking down your thought process. Ensure that each reasoning step is explicitly connected to observable details in the image or text, and articulate your conclusion in a clear and logical manner.

Chain-of-thought reasoning tasks require a more complex interaction between visual and textual information. The prompt encourages users to break down their reasoning process into clear, logical steps, each supported by specific details from the image or text.

Furthermore, we explore three categories of instructions to enhance the MM-ICL process:

**Introductory Instruction** ($\mathcal{I}_{intro}$)  This instruction provides an overview of the task before presenting any demonstrations. As depicted in Figure 4 (a), the introductory instruction $\mathcal{I}_{intro}$ is positioned at the beginning of the ordered demonstration list $\mathcal{L}$. This setup aims to set the context for the subsequent examples. Specifically, the overall prompt template is as follows:

> *<Instruction $\mathcal{I}_{intro}$>*
> **[DEMONSTRATIONS]**
>  **[REQUEST]** % Shot 1
>    *<Visual Input $\mathcal{I}_1^{vis}$>*   *<Textual Input $\mathcal{I}_1^{txt}$>*
>  **[RESPONSE]**
>    *<Textual Output $\mathcal{I}_1^{vis}$>*
> . . .
> **[QUERY]**
>  **[REQUEST]** % User Query
>    *<Visual Input $\mathcal{I}_q^{vis}$>*   *<Textual Input $\mathcal{I}_q^{txt}$>*

**Summative Instruction** ($\mathcal{I}_{sum}$)  This instruction offers a summary after the examples, guiding the model to apply the learned concepts to real-world problems. As shown in Figure 4 (b), the summative instruction $\mathcal{I}$ is added at the end of the demonstration list $\mathcal{L}$. This helps in reinforcing the learning objectives and expected outcomes. Specifically, the overall prompt template is as follows:

> *<Instruction $\mathcal{I}_{intro}$>*
> **[DEMONSTRATIONS]**
>  **[REQUEST]** % Shot 1
>    *<Visual Input $\mathcal{I}_1^{vis}$>*   *<Textual Input $\mathcal{I}_1^{txt}$>*
>  **[RESPONSE]**
>    *<Textual Output $\mathcal{I}_1^{vis}$>*
> . . .
> In summary, *<Instruction $\mathcal{I}_{sum}$>*
> **[QUERY]**
>  **[REQUEST]** % User Query
>    *<Visual Input $\mathcal{I}_q^{vis}$>*   *<Textual Input $\mathcal{I}_q^{txt}$>*

**Intra-demonstration Instruction** ($\mathcal{I}_{intra}$)  This instruction embeds task instructions within each example, assisting the model in understanding the task requirements during the learning process. As

| Model | OKVQA [Marino et al., 2019] | | | VQA-v2 [Goyal et al., 2017] | | |
|---|---|---|---|---|---|---|
| | Accuracy | BERTScore | Token F1 | Accuracy | BERTScore | Token F1 |
| OpenFlamingo [Awadalla et al., 2023] | 40.28 | 78.10 | 17.45 | 53.33 | 83.34 | 25.67 |
| GPT4V [OpenAI: et al., 2023] | 54.28 | 85.97 | 25.23 | 69.69 | 84.89 | 29.18 |
| IDEFICS2 [Laurençon et al., 2024b] | 55.32 | 87.61 | 27.81 | 71.28 | 87.98 | 35.46 |

Table 3: The correlation analysis of the indicators and reproduced accuracy. The results are obtained by testing on a subset of the test set.

illustrated in Figure 4 (c), the intra-demonstration instruction $\mathcal{I}$ is included within each demonstration $x_i$ in the list $\mathcal{L}$. This method ensures that the task instructions are continuously reinforced throughout the learning process. Specifically, the overall prompt template is as follows:

---

**[DEMONSTRATIONS]**

**[REQUEST]** % Shot 1

  *<Visual Input $\mathcal{I}_1^{vis}$>*   *<Textual Input $\mathcal{I}_1^{txt}$> <Instruction $\mathcal{I}_{intra}$>*

**[RESPONSE]**

  *<Textual Output $\mathcal{I}_1^{vis}$>*

. . .

**[QUERY]**

**[REQUEST]** % User Query

  *<Visual Input $\mathcal{I}_q^{vis}$>*   *<Textual Input $\mathcal{I}_q^{txt}$> <Instruction $\mathcal{I}_{intra}$>*

---

By systematically incorporating these instruction categories into the MM-ICL framework, we aim to investigate their impact on model performance and task comprehension.

## D  Prompt Robust

In our preliminary experiments, we observed that variations in prompts do not significantly alter the overall conclusions. Specifically, we employed multiple prompts—differing in instructions and delimiters—while maintaining equivalent semantic content but varying linguistic expression. As demonstrated in Table 4, the influence of these different prompts on the results is minimal. This suggests that our findings are robust to changes in prompt formulation, thereby supporting the reliability of the experimental outcomes.

| | Caption | | | VQA | Classification | | Reasoning | | AVG |
|---|---|---|---|---|---|---|---|---|---|
| | CIDER | BERTScore | Token F1 | BERTScore | Acc | F1 | Acc | RAS | |
| P1 | 12.03 | 85.85 | 22.53 | 86.67 | 59.93 | 54.62 | 59.52 | 92.04 | 59.15 |
| P2 | 14.01 | 86.77 | 23.59 | 86.00 | 58.53 | 53.61 | 61.85 | 91.86 | 59.53 |
| P3 | 13.91 | 86.92 | 24.70 | 87.63 | 59.74 | 52.14 | 61.89 | 93.05 | 60.00 |
| P4 | 14.44 | 86.48 | 23.14 | 87.77 | 60.23 | 50.48 | 60.54 | 92.27 | 59.42 |

Table 4: Performance across different prompts (i.e., P1, P2, P3 and P4).

