# OpenReview forum: "What Factors Affect Multi-Modal In-Context Learning? An In-Depth Exploration"
_NeurIPS.cc/2024/Conference — NeurIPS 2024 poster_

### Official Review · Reviewer_fN5J · 2024-06-21

**Soundness:** 1
**Presentation:** 2
**Contribution:** 2
**Rating:** 4
**Confidence:** 4

**Summary:**

The paper studies what influences multimodal ICL and finds that using multimodal retrievers to find demonstrations, ordering of modalities in the context and introductory instructions help to improve the few-shot performance. The study spans 6 models and several multimodal datasets.

**Strengths:**

- The paper addresses an important problem. Multimodal ICL is little explored.
- The paper conducts experiments spanning several models and datasets.

**Weaknesses:**

1. The paper claims to be the first to study multimodal ICL. However, previous works [1,2] conducted very similar studies and there is no mention for these previous works.
2. The paper does not consider standard benchmarks. In addition it uses metrics not generally used to report scores in other papers. For example, the performance on VQAv2 is measured with Accuracy not BERTscore. I think the considered benchmarks and metrics are leading to false conclusions as detailed later, either because these metrics/benchmarks are flawed/limited or just saturated. To support the paper claims it is important to consider a typical evaluation setup. For example on benchmarks such as VQAv2, COCO captioning, NoCaps, TextCaps, OK-VQA, TextVQA ..  using typical scores such as Accuracy, CIDEr/BLEU…
3. Related to previous points. Table 1 shows that IDEFICS 8B is comparable to GPT4V ! (e.g. VQA BertScore 85.42 GPT4V vs 86.11 IDEFICS few-shot random) IDEFICS 8B is significantly worse on almost all multimodal benchmarks. I am not sure at this point how much I can rely on the paper results and findings.
4. I have some issues with some claims that I don’t find consistent with previous studies. These claims should be refined or supported with more evidence:
- “The number of demonstrations does not significantly impact MM-ICL”: usually the more shots the better the performance [3,4,5] and the improvement is significant.
- “increasing model parameters from 8 billion to over 100 billion does not primarily drive performance improvements, suggesting that multi-modal context understanding and alignment are more crucial for MM-ICL than model scaling”: Other works clearly show that increasing model size increases ICL performance [3,5]. To make such claim, only the scale of the model should change, but in Table 1 the authors seems to compare different models (e.g. GPT4-V vs Qwen-VL 10B)

5. Wrong citations: IDEFICS 8B is not done by Awadallah et al (Table 1) !

6.  The paper figures are not properly explained:
- What the colors means in Fig 10?
- Which scores are you showing in Fig 8 and 9?
Please write detailed captions to explain the paper figures.

7. What is BELU score? Is it the same as BLEU?



References:

[1] Baldassini, Folco Bertini, et al. "What Makes Multimodal In-Context Learning Work?." Proceedings of the IEEE/CVF Conference on Computer Vision and Pattern Recognition. 2024.

[2] Chen, Shuo, et al. "Understanding and Improving In-Context Learning on Vision-language Models." ICLR 2024 Workshop on Mathematical and Empirical Understanding of Foundation Models (2024).

[3] Alayrac, Jean-Baptiste, et al. "Flamingo: a visual language model for few-shot learning." Advances in neural information processing systems 35 (2022): 23716-23736.

[4] Laurençon, Hugo, et al. "Obelics: An open web-scale filtered dataset of interleaved image-text documents." Advances in Neural Information Processing Systems 36 (2024).

[5] Shukor, Mustafa, et al. "Beyond task performance: evaluating and reducing the flaws of large multimodal models with in-context-learning." The Twelfth International Conference on Learning Representations. 2023.

**Questions:**

Check weaknesses (e.g. 6. 7)

**Limitations:**

Few limitations are discussed in the paper.

---

> ### Author Rebuttal · Authors · 2024-08-05
>
> We sincerely appreciate your thorough and insightful comments on our work. In the following, **we will clarify your concerns and we would greatly appreciate it if you can reconsider the work in light of our clarification**.
>
> ---
> **Q1:** The difference between our work with previous studies [1,2].
>
> **R1:** Thanks for your insightful feedback. We sincerely clarify the main differences from two aspects:
> 1. **Unified Perspective.** Our work makes the first attempt to unify the current multi-modal In-context learning (MM-ICL) process, which takes a meaningful step to offer a unified view and guideline for researchers to build better MM-ICL. Previous works [1,2] can also be integrated in our perspective.
> 2. **Comprehensive Investigation.** According to the unified perspective, we conduct a comprehensive exploration and provide insightful observations for each process within the unified perspective. However, previous studies [1,2] ***mainly focus on exploring the sample representation subpart of the retrieval process***, which is only a subset of our exploration.
>
> More intuitive comparisons can be found in Figure 1 of the supplementary material. We will follow your suggestions to add a detailed discussion in the next version.
>
> [1] Baldassini et al. What Makes Multimodal In-Context Learning Work? CVPR 2024 Workshop.
>
> [2] Chen et al. Understanding and Improving In-Context Learning on Vision-language Models. ICLR 2024 Workshop.
>
> ---
> **Q2:** The paper does not consider standard benchmarks and metrics.
>
> **R2:** Thanks for your constructive comments and we will answer your concerns point by point.
> - **Benchmark Concern:** We use the standard benchmark M3IT [1], which consists of a series of common datasets divided into four categories. Detailed information is shown in Table 1. We will add more data descriptions in the next version.
> - **Metric Concern:** In our experiment, we use the standard CIDEr metric for image caption task. However, since M3IT includes various VQA tasks with many free-form answers, the Acc. metric is not suitable. Therefore, inspired by the success of free-form and precise answer hybrid evaluation in machine reading comprehension, we use the Token F1 and BERTScore as metrics for evaluating semantics and exact keyword accuracy. We will add more discussion in the next version.
>
> |Dataset|Data Class|
> |:--:|:--:|
> |COCO Caption|IC|
> |TextCap|IC|
> |Paragraph Captioning|IC|
> |COCO Text|CLS|
> |ImageNet Image Classification|CLS|
> |IQA|CLS|
> |Image-Text Matching|CLS|
> |e-SNLI-VE|CLS|
> |Multi-modal Fact Checking|CLS|
> |VQA v2|VQA|
> |DocVQA|VQA|
> |OCR-VQA|VQA|
> |ST-VQA|VQA|
> |Text-VQA|VQA|
> |GQA|VQA|
> |OKVQA|VQA|
> |A-OKVQA|VQA|
> |ScienceQA|R|
> |M3CoT|R|
> ||
>
> Table 1: Dataset in M3IT, where IC: Image Captioning, CLS: Classification, VQA: Visual Question Answering, R: Reasoning (with NL rationale).
>
> [1] Li et al. M$^3$IT: A Large-Scale Dataset towards Multi-Modal Multilingual Instruction Tuning. Arxiv 2024.
>
> ---
> **Q3:** Table 1 shows that IDEFICS 8B is comparable to GPT4V! However, IDEFICS 8B is worse on almost all benchmarks. Why？
>
> **R3:** Thanks for your insightful comment. Sorry for confusing you. We totally agree that IDEFICS 8B is worse on almost all benchmarks. However, in our work, Table 1 shows that the results of IDEFICS2 8B [1], not IDEFICS 8B. In fact, IDEFICS2 is comparable with GPT4V, as shown in the following Table 2. We will add more discussion in the next version to make it clearer.
>
> ||MathVista|MMBench|
> |:--:|:--:|:--:|
> |GPT4V|49.9[2]|74.3[3]|
> |IDEFICES2-8B|51.6[1]|76.8[1]|
> ||
>
> Table 2: Performance comparison.
>
> [1] Laurençon et al. What matters when building vision-language models? Arxiv 2024
>
> [2] Lu et al. MathVista: Evaluating Math Reasoning in Visual Contexts. ICLR 2024
>
> [3] Liu et al. MMBench: Is Your Multi-modal Model an All-around Player? Arxiv 2023
>
> ---
> **Q4:** Some claims should be refined or supported with more evidence:
> 1. Other works usually show the more shots the better the performance.
> 2. Other works show that increasing model size increases ICL performance.
>
> **R4:** Thanks for your valuable suggestions. We will answer your questions point by point.
> - **Response to (1):** Yes, your understanding is correct. In fact, previous works [1-3] concluded that 'more shots lead to better performance' mainly for the VQA task.
> We also observed this trend in VQA task. But in other tasks like multi-modal chain-of-thought reasoning, it can even worsen results with more demonstrations [4]. In the future, we will follow your suggestion and provide more fine-grained conclusions, such as offering fine-grained observations for different tasks.
> - **Response to (2):** We totally agree with you. With the same model, a larger model generally performs ICL better. In our work, we were surprised to find that IDEFICS2 (8B) outperformed GPT-4V (>100B) on some tasks, indicating that beyond parameter size, multi-modal context understanding and alignment are also crucial for MM-ICL. We will refine our conclusion in the next version according to your suggestion.
>
> [1] Alayrac et al. Flamingo: a visual language model for few-shot learning. NeurIPS 2022.
>
> [2] Laurençon et al. Obelics: An open web-scale filtered dataset of interleaved image-text documents. NeurIPS 2024.
>
> [3] Shukor et al. Beyond task performance: evaluating and reducing the flaws of large multimodal models with in-context-learning. ICLR 2023.
>
> [4] Chen et al. M3CoT: A Novel Benchmark for Multi-Domain Multi-step Multi-modal Chain-of-Thought. ACL 2024.
>
> ---
> **Q5:** Other writing typos and tips (e.g., wrong citations; paper figures are not properly explained)
>
> **R5:** Thanks for your valuable feedback. We will follow your suggestions to thoroughly fix wrong citations and add detailed figure captions. Specifically, different colors denote different models in Fig 10. The scores in Fig 8 and 9 represent the Average score (AVG).
>
> ---
> **Q6:** What is BELU score?
>
> **R6:** Sorry for the confusion. It is BLEU. We will polish our work in the next version.

---

> > ### Author Response · Authors · 2024-08-10
> > **Thanks for your time and effort**
> >
> > Thank you very much for your time and valuable feedback. Hope our clarifications can address your concerns, and we sincerely hope that you can reconsider our work in light of these clarifications. If you have any further comments, please do not hesitate to contact us. We greatly appreciate your selfless contributions to the community.

---

> > ### Comment · Reviewer_fN5J · 2024-08-10
> >
> > Thanks for the detailed response. My main concerns about this submission is the evaluation. Without proper evaluation no serious conclusions can be drawn. I suggest to evaluate the model on a "broad" range on benchmarks "commonly" used by the multimodal community. Seeing the evaluation and conclusions, it seems that using the  M3IT benchmark is not a good choice. Showing 2 scores of IDEFICS2 that are worse but close to GPT4V is not enough to conclude that these 2 models are even comparable (the authors can check the evaluation of these models on broad range of tasks). I highly encourage the authors to work on the evaluation and avoid any claim that goes against previous serious works, unless a "rigorous, extensive and convincing" arguments/experiments support that.  Also regarding the originality, the work is "among" the first and not "the first". I think the paper needs additional work on the claims and experiments to be accepted at top venues. Given the importance of the studied topic, and the authors promises to refine and clarify some claims I will not decrease my score.

---

> ### Author Response · Authors · 2024-08-11
> **Clarification for Your Feedback**
>
> Thanks for your timely feedback. We would like to further clarify your concerns.
>
> **Q1:** Seeing the evaluation and conclusions, it seems that using the M3IT benchmark is not a good choice.
>
> **R1:** Thanks for your suggestion. In our detailed response R2 in previous rebuttal, M3IT includes four classic types of tasks that **contain almost all of your mentioned datasets** (e.g., VQAv2, COCO captioning, TextCaps, OK-VQA, TextVQA). We sincerely believe that these cover a broad range of categories for evaluation. We will add more details in the next version.
>
> ---
> **Q2:** Showing 2 scores of IDEFICS2 that are worse but close to GPT4V is not enough to conclude that these 2 models are even comparable.
>
> **R2:** Thanks for your comment. We sincerely believe that there are some misunderstandings. In fact, in some previous findings [1,2,3], **IDEFICS2 even outperforms GPT4V by 2 scores** (see the Table below). As shown in Table 1 in the submitted paper, the performance of these two models are also comparable. The core difference lies in the image captioning task. Nevertheless, the metrics you mentioned, such as CIDEr, are inappropriate for evaluating open-ended image caption. Human judgments of alignment often differ significantly because GPT-4V tends to provide more detailed responses, whereas IDEFICS2 prefers shorter ones. However, in most image captioning datasets, the golden captions are brief, which presents challenges for traditional metrics [4]. Therefore, such observations motivate us to explore semantic-level BERTScore metric in addition to CIDER for open-ended image caption task.
>
> ||MathVista|MMBench|
> |:--:|:--:|:--:|
> |GPT4V |49.9[2]|74.3[3] |
> |IDEFICES2-8B|51.6[1] |76.8[1] |
> Table: Performance comparison.
>
> [1] Laurençon et al. What matters when building vision-language models? Arxiv 2024
>
> [2] Lu et al. MathVista: Evaluating Math Reasoning in Visual Contexts. ICLR 2024
>
> [3] Liu et al. MMBench: Is Your Multi-modal Model an All-around Player? Arxiv 2023
>
> [4] What you see is what you read? improving text-image alignment evaluation. NeurIPS2023
>
> ---
> **Q3:** I highly encourage the authors to work on the evaluation and avoid any claim that goes against previous serious works.
>
> **R3:** Thanks for your constructive comment. Actually, **our findings are not in conflict with previous work**. For example, our findings in multi-modal chain-of-thought reasoning that simply adding more shots does not always lead to improved performance, which is consistent with the previous work [1]. We think that the following might be reasons why more demonstrations may not improve performance in these tasks, which is acknowledged by Reviewer #Wrbn.
>
> 1.	**Cognitive Overload**: For complex tasks, understanding complex demonstrations is difficult. More demonstrations can overwhelm the model, making it harder to process and integrate information effectively.
> 2.	**Complexity of Reasoning Tasks**: We observe that for reasoning tasks, the performance improvement brought by the number of demonstrations is not even as good as using different retrievers. It shows that reasoning tasks require sophisticated integration of information, where quality trumps quantity.
>
> In addition, in our experiments, we also observe that more shots lead to better performance in VQA task, which also align with [2][3][4].  We will add more discussion in the next version.
>
> [1] Chen et al. M3CoT: A Novel Benchmark for Multi-Domain Multi-step Multi-modal Chain-of-Thought. ACL 2024.
>
> [2] Alayrac et al. Flamingo: a visual language model for few-shot learning. NeurIPS 2022.
>
> [3] Laurençon et al. Obelics: An open web-scale filtered dataset of interleaved image-text documents. NeurIPS 2024.
>
> [4] Shukor et al. Beyond task performance: evaluating and reducing the flaws of large multimodal models with in-context-learning. ICLR 2023.
>
> ---
> **Q4:** Regarding the originality
>
> **R4:** Thanks for your kind mention. We sincerely think that our work is the first to unify the current multi-modal In-context learning (MM-ICL) process and conduct a comprehensive exploration for each process within the unified perspective. The provided findings in this work are meaningful and interesting, **which have been recognized by Reviewer #Pm2y, Reviewer #Wrbn, Reviewer #jbhA**.
>
> - Multi-modal alignment is the bottleneck for MM-ICL.
> - Intra-demonstration ordering holds greater importance than inter-demonstration ordering.
> - Introductory instruction guides better task understanding for MM-ICL.
>
> In addition, when the mentioned work [1] is presented at CVPR Workshop, the NeurIPS deadline has already passed.
>
> [1] Baldassini et al. What Makes Multimodal In-Context Learning Work? CVPR 2024 Workshop.
>
> **We greatly appreciate the time you've invested in reviewing our response**. We hope the further clarification can address your concerns and **we sincerely hope that you can reconsider your score in light of our clarifications**. We **promise** to incorporate your all suggestions to improve our work.

---

> > ### Comment · Reviewer_fN5J · 2024-08-11
> >
> > Thanks again for the detailed response.
> >
> > To clarify, my concerns about the evaluation is related mainly to the metrics and in particular for VQA tasks. I think VQA tasks are very important as they cover the main use cases of such models. The accuracy is what mainly reported on VQA tasks as seen in tons of recent papers.
> >
> > - Does F1/BertScore correlate with the accuracy?
> > - If the authors think that the accuracy is not suitable here they should justify why and it needs a separate study to shows this (which I don't think is within the scope of this paper).
> >
> > It is very likely that if the authors report the accuracy on VQA tasks they will see different observations (e.g. the significant gap between GPT4V and IDEFICS 2, for example GPT4V [1] (zero-shot) got 56.8 compared to 43.5 for IDEFICS 2 on MMMU benchmark).
> >
> > It is possible to find some benchmarks where weaker models have close scores to much stronger ones. This might be due to many reasons, some of them could be the benchmark/metrics are biased/flawed/saturated or do not reflect the real model capabilities, the weaker model is finetuned on the training set or similar datasets to the evaluated benchmarks ...
> >
> > **Nonetheless for the purpose of this paper and similar papers that conduct analysis to understand these models, the benchmarks/metrics should be well chosen to properly draw conclusions.**
> >
> > My worries are if the paper is properly evaluated, different conclusions might be drawn. If the authors can show that for example the VQA accuracy on several VQA datasets is correlated with F1/BertScore or does not change the main paper messages, I might consider increasing my score.

---

> ### Author Response · Authors · 2024-08-12
> **Gratitude for Your Detailed Feedback**
>
> We would like to extend our sincere gratitude for your thoughtful and detailed feedback. In addition, we are very grateful for the opportunity to further clarify your worry. We notice that your main concern is “Does F1/BertScore correlate with the accuracy?” and we sincerely agree with your concern. In the following, we will do our best to address your concern.
>
> **Q:** Does F1/BertScore correlate with the accuracy?
>
> **A:** Thanks for your insightful feedback. The answer is “**Yes**”. According to your suggestion, we have adopted accuracy for evaluating several VQA datasets including OK-VQA and VQAv2. The results are shown in Table 1 and Table 2 below:
>
> |Model| Acc. | BERTScore | Token F1 |
> |:--:|:--:|:--:|:--:|
> |OpenFlamingo | 40.28 | 78.10 | 17.45 |
> |GPT4V | 54.28 | 85.97 | 25.23 |
> |IDEFICS2 | 55.32 | 87.61 | 27.81 |
> Table 1: Performance on OKVQA.
>
> |Model| Acc. | BERTScore | Token F1 |
> |:--:|:--:|:--:|:--:|
> |OpenFlamingo | 53.33 | 83.34 | 25.67 |
> |IDEFICS2| 69.69 | 84.89 | 29.18 |
> |GPT4V| 71.28 | 87.98 | 35.46 |
> Table 2: Performance on VQAv2.
>
> From the results, we observe that accuracy is positively correlated with BERTScore and F1. We attribute it to the fact that higher semantic relevance and exact keyword performance can lead to higher accuracy. Such observation also demonstrates the main paper messages cannot be changed. We sincerely believe that the findings in our work can directly contribute to the MM-ICL community. **In the next version, we promise to add more experiments and discussions to enrich our work**.
>
> Your insights and suggestions can significantly contribute to enhancing the quality and clarity of the work. **We are very encouraged that the further clarifications can address your concern**. Thank you once again for your selfless contributions to the community.

---

> > ### Comment · Reviewer_fN5J · 2024-08-13
> >
> > Thanks for your response.
> >
> > It seems that the accuracy is correlated with the metrics used in the paper (despite the fact that this needs more elaborated experiments), and better evaluation is less likely to change the main messages of the paper.
> >
> > However, I believe the paper should consider the benchmarks and metrics commonly used to compare recent  models to draw its conclusions, and can be improved on this side. Also, the authors shouldn't include any scores that raise suspicions about the work, I again ask the authors to reconsider the GPT4V scores (this model is significantly better than IDEFICS2).
> >
> > Given the authors clarifications and other reviewers responses, and to encourage the authors to include what is discussed here, as well as other reviewers responses, I will increase my score from 3 to 4.
> >
> > Note: the VQAv2 score of GPT4V is 77.2 not 71.28 (Table 4 in [1])
> >
> > [1] McKinzie, Brandon, et al. "Mm1: Methods, analysis & insights from multimodal llm pre-training." arXiv preprint arXiv:2403.09611 (2024).

---

> > > ### Author Response · Authors · 2024-08-13
> > > **Thanks Again for Your Time and Effort**
> > >
> > > We are writing to express our heartfelt gratitude for your valuable feedback. Your insightful and valuable suggestions can significantly contribute to enhancing the solidity of our work. We will follow your suggestions to polish our work in the next version.
> > >
> > > Thank you once again for your valuable contributions.

---

### Official Review · Reviewer_jbhA · 2024-07-01

**Soundness:** 3
**Presentation:** 3
**Contribution:** 3
**Rating:** 8
**Confidence:** 5

**Summary:**

This work explores an interesting research question: “What factors affect the performance of MM-ICL?” To this end, they conduct comprehensive experiments on the three fundamental steps of MM-ICL: demonstration retrieval, demonstration ordering, and prompt construction. In addition, they explore 20 strategies across 4 tasks with 6 representative vision large language models (VLLMs).

**Strengths:**

（1）This work explores the research question “What factors affect the performance of MM-ICL?” which is an important topic in MM-ICL literature.

（2）This work observes some interesting findings, including (i) multi-modal alignment is the bottleneck for MM-ICL; (ii) intra-demonstration ordering holds greater importance than inter-demonstration ordering; (iii) introductory instruction guides better task understanding for MM-ICL, which can offer a unified view and guideline for researchers to build better MM-ICL.

（3）This work conducts very detailed experiments by exploring 20 strategies across 4 tasks with 6 representative vision large language models (VLLMs), which is impressive and encouraging.

**Weaknesses:**

Overall, I have no major issues. There are a few points that can be improved:
(1) The 'Exploration of MM-ICL Prompt Construction' section can be explained more clearly.

(2) Some parts of the appendix, such as the implementation details of the baseline, can be moved to the main text.

(3) The 'Limitations' section can include some specific limitations related to this work.

**Questions:**

See above comments.

**Limitations:**

See above comments.

---

> ### Author Rebuttal · Authors · 2024-08-05
>
> Thanks for your acknowledgment and interest in our work! We sincerely appreciate your thorough and insightful comments on our work, and we will address each of your main concerns below:
>
> ---
> **Q1:** The 'Exploration of MM-ICL Prompt Construction' section can be explained more clearly.
>
> **R1:** Thanks for your constructive suggestion. We will follow your suggestion to explain more details about the 'Exploration of MM-ICL Prompt Construction' section, including providing more description in Figure 4 and adding more details in three instruction categories explanation.
>
> ---
> **Q2:** Some parts of the appendix, such as the implementation details of the baseline, can be moved to the main text.
>
> **R2:** Thanks for your insightful feedback. We totally agree with your point. We will add the implementation details of the baseline to the main text in the next version.
>
> ---
> **Q3:** The 'Limitations' section can include some specific limitations related to this work.
>
> **R3:** Thanks for your insightful suggestion. We will discuss more limitations in the next version. For example, one of limitations in our work lies in the mis-consideration of some image instructions, such as grounding and adding extra arrows. These may require more sophisticated human design and are not supported by most current models.

---

> > ### Comment · Reviewer_jbhA · 2024-08-13
> >
> > Thanks for your response. I read this work again and went through the author's responses to other reviewers. I think the quality and contributions of this paper are solid and clear. The topic studied is important. Although some progress has been made in text-only In-context Learning, comprehensive investigation of multi-modal In-context Learning remains underexplored. The findings and practices in this paper, including the design of demonstration ordering and instruction prompts, can provide valuable insights for future research. I am inclined to raise my score to 8 and champion this paper.

---

> > > ### Author Response · Authors · 2024-08-13
> > > **Gratitude for Your Detailed Feedback**
> > >
> > > We sincerely thank you for investing your time to review our work. We are encouraged by your acknowledgment and interest in our work. We will incorporate your suggestions to polish our work in the next version. Thank you once again for your valuable contributions.

---

### Official Review · Reviewer_Wrbn · 2024-07-09

**Soundness:** 4
**Presentation:** 3
**Contribution:** 3
**Rating:** 7
**Confidence:** 4

**Summary:**

The work presents an in-depth analysis of the factors influencing the performance of Multi-modal In-Context Learning (MM-ICL). The authors systematically investigate the core steps of MM-ICL, including demonstration retrieval, ordering, and prompt construction. Utilizing six vision large language models and a variety of strategies, the study uncovers the significance of multi-modal alignment, the importance of intra-demonstration ordering, and the role of introductory instructions in enhancing task comprehension.

**Strengths:**

1. Comprehensive Analysis: The paper offers a thorough examination of the factors affecting MM-ICL, which is a significant contribution to the field.
2. Clear Structure: The paper is well-organized, making it easy to follow the research question, methodology, findings, and conclusions.
3. Potential Impact: The findings in this work can provide a foundational guide for optimizing MM-ICL strategies in future research.

**Weaknesses:**

1. The captions in the paper can be enriched to help readers gain a better understanding.
2. In Section 5.3, can you provide more analysis on why the number of demonstrations does not significantly impact MM-ICL?
3. The experimental section needs to be supplemented with a description of the LLM backbone.

**Questions:**

1. In Section 5.3, can you provide more analysis on why the number of demonstrations does not significantly impact MM-ICL?
2. In your experiments, how did you handle multiple image inputs?
3. Will different prompts affect the exploration experimental results?

**Limitations:**

This work presented some limitations in the submitted paper including: (1) extending the exploration to video modal ICL and (2) multi-lingual multi-modal ICL scenarios.

---

> ### Author Rebuttal · Authors · 2024-08-05
>
> Thanks for your acknowledgment and interest in our work! We sincerely appreciate your thorough and insightful comments on our work, and we will address each of your main concerns below:
>
> ---
> **Q1:** The captions in the paper can be enriched to help readers gain a better understanding.
>
> **R1:** Thanks for your constructive suggestion. We will follow your suggestion to add more detailed captions in the next version.
>
> ---
> **Q2:** In Section 5.3, can you provide more analysis on why the number of demonstrations does not significantly impact MM-ICL?
>
> **R2:** Thanks for your insightful feedback. We find that in complex reasoning tasks such as multi-step multi-modal chain-of-thought reasoning, more demonstrations will not lead to better performance, which is consistent with the observation [1]. We think that the following might be reasons why more demonstrations may not improve performance in these tasks:
> 1. **Cognitive Overload:** For complex tasks, understanding complex demonstrations is difficult. More demonstrations can overwhelm the model, making it harder to process and integrate information effectively.
> 2. **Complexity of Reasoning Tasks:** We observe that for reasoning tasks, the performance improvement brought by the number of demonstrations is not even as good as using different retrievers. It shows that reasoning tasks require sophisticated integration of information, where quality trumps quantity.
>
> We will add more discussion in the next version.
>
> [1] Chen et al. M3CoT: A Novel Benchmark for Multi-Domain Multi-step Multi-modal Chain-of-Thought. ACL 2024.
>
> ---
> **Q3:** The experimental section needs to be supplemented with a description of the LLM backbone.
>
> **R3:** Thanks for your constructive suggestion. We will add a detailed description of each LLM backbone used in the next version according to your suggestion.
>
> ---
> **Q4:** In your experiments, how did you handle multiple image inputs?
>
> **R4:** Thanks for your insightful comment. There are two main categories for handling multiple image inputs:
> - **Tokenizer-based LVLM:** It supports directly tokenizing the images and then directly concating them with textual tokens in an interlaced sequence.
> - **Encoding-based LVLM:** The soft encoding of images is concated at the embedding layer to form an interlaced sequence of images and text.
>
> ---
> **Q5:** Will different prompts affect the exploration experimental results?
>
> **R5:** Thank you for your insightful feedback. In our preliminary experiments, we found that different prompts do not affect the overall conclusions. For example, we used different prompts (including instructions and delimiters) with the same semantics but different expressions. The results are shown in Table 2 below, and it can be seen that the impact of different prompts is not very large. We will add more discussion in the next version.
>
> |      | Caption | Caption | VQA | VQA | Classification | Classification | Reasoning | Reasoning | |
> | :--: | :--: | :--: | :--: | :--: | :--: | :--: | :--: | :--: | :--: |
> | | CIDER | BERTScore | Token F1 | BERTScore | Acc | F1 | Accuracy | RAS | AVG |
> | P1 | 12.03 | 85.85 | 22.53 | 86.67 | 59.93 | 54.62 | 59.52 | 92.04 | 59.15 |
> | P2 | 14.01 | 86.77 | 23.59 | 86.00 | 58.53 | 53.61 | 61.85 | 91.86 | 59.53 |
> | P3 | 13.91  | 86.92  | 24.70 | 87.63 | 59.74 | 52.14 | 61.89  | 93.05 | 60.00 |
> | P4 | 14.44 | 86.48 | 23.14 | 87.77 | 60.23 | 50.48 | 60.54 | 92.27 | 59.42 |
> ||
>
> Table 2: Performance across different prompts.

---

> > ### Comment · Reviewer_Wrbn · 2024-08-09
> > **Concerns Addressed**
> >
> > Thanks the authors for your clarifications, which I think possibly have addressed all my previous concerns. I like the insightful explanation about demonstrations for complex reasoning tasks, including Cognitive Overload and the Complexity of Reasoning Tasks. Actually, I’m working on MM-ICL for reasoning, and from my practice, for some complex reasoning tasks, providing just a few demonstrations may not be sufficient. So it is reasonable and necessary to explicitly construct some clues from multimodal reasoning examples when creating demonstrations. Overall, I think the findings authors give are meaningful and interesting, and believe they will show more impact in MM-ICL or relevant topics. One tip for authors: as future work, I think it would also be interesting to explore a unified theoretical framework for MM-ICL.

---

> > > ### Author Response · Authors · 2024-08-09
> > > **Gratitude for Constructive Feedback**
> > >
> > > We sincerely thank you for investing your time to review our response. Your insightful and valuable suggestions have significantly contributed to enhancing the solidity of our work. We will follow your suggestions to enhance our work in the next version.
> > >
> > > Thank you once again for your valuable contributions.

---

### Official Review · Reviewer_Pm2y · 2024-07-13

**Soundness:** 3
**Presentation:** 3
**Contribution:** 2
**Rating:** 5
**Confidence:** 3

**Summary:**

The paper investigates the underlying factors that influence the effectiveness of Multi-Modal In-Context Learning (MM-ICL). The authors conducted extensive experiments on three core steps of MM-ICL: demonstration retrieval, demonstration ordering, and prompt construction using six vision large language models and 20 strategies. Their findings highlight the necessity of a multi-modal retriever for demonstration retrieval, the importance of intra-demonstration ordering, and the enhancement of task comprehension through introductory instructions in prompts. This study aims to provide a foundational guide for optimizing MM-ICL strategies in future research.

**Strengths:**

The article is well-written, and the authors have thoroughly addressed several important concepts that reviewers might raise. The paper effectively covers a broad range of factors affecting MM-ICL. The authors have conducted experiments across multiple models and strategies, providing a diverse set of data points, which is valuable for identifying trends and patterns.

**Weaknesses:**

Some weaknesses are listed below:

1. The results of the experiments seem apparent and intuitive. For example, the multi-modal retriever performs better than the single modality retriever because it incorporates more information. Additionally, intra-demonstration ordering is crucial as it affects the structure of the input sample and how the model processes the input. As a result, the findings do not seem particularly inspiring.

2. Some qualitative analysis is missing. It would be beneficial to compare how different in-context examples affect the answers.

3. Considering that the margin of improvement in some metrics is quite small, certain designs of in-context examples may not be as important. For instance, as shown in Table 1, in most scenarios, a single textual retriever is sufficient. Therefore, using a multi-modal retriever, which may consume more computational resources, is unnecessary.

4.	Is it appropriate to combine all tasks to provide a universal paradigm for constructing MM-ICL strategies? For example, in classification tasks, the multi-modal retriever does not seem to perform the best.

There are also some minor mistakes in the paper:

1. It is more common to use the term "Large Vision-Language Models (LVLMs)" instead of "Vision Large Language Models (VLLMs)," which you use throughout the paper (e.g., line 43).

2.The reference for the model Qwen-VL in Table 1 is incorrect; the model is not from OpenAI.

3.In Table 1, the Otter Model results are not highlighted in bold.

**Questions:**

In addition to the list of weaknesses above, here are some questions for the authors to address:

1.For multi-modal demonstration retrieval, why do you use a multi-modal encoder to calculate the similarity instead of using the product of text and vision single modality similarity? Exploring this alternative approach might offer additional insights into multi-modal retrieval.

2.For the OpenAI GPT-4V model, how do you reconstruct the order of the text and the image? As far as I know, the API does not provide a way to structure the order of text and images.

**Limitations:**

Limitations are discussed in the paper. and the authors do not foresee any negative societal impact.

---

> ### Author Rebuttal · Authors · 2024-08-05
>
> Thanks for your acknowledgment and interest in our work! We sincerely appreciate your thorough and insightful comments on our work, and we will address each of your main concerns below:
>
> ---
> **Q1:** The results of the experiments seem apparent and intuitive.
>
> **R1:** Thanks for your constructive feedback. We sincerely believe that our work is a meaningful step in the systematic exploration of the MM-ICL and provides some insightful observations.
>
> 1. Previous studies mainly considered the inter-demonstration ordering. To our knowledge, we are the first to explore the intra-demonstration ordering and find that this ordering is significantly more important than inter-demonstration ordering, hoping to provide insightful guidance in the future.
> 2. Additionally, we thoroughly explored how to effectively construct MM-ICL prompts by investigating Introductory Instruction, Summative Instruction, and Intra-demonstration Instruction, which is less explored in the previous research.
> 3. In addition, we provide more insights on how to insert separators, optimize domain selection, and refine distance metrics, offering guidance on best practices for constructing MM-ICL.
>
> We will follow your suggestion to add more discussion in the next version.
>
> ---
> **Q2:** Some qualitative analysis is missing. It would be beneficial to compare how different in-context examples affect the answers.
>
> **R2:** Thanks for your insightful suggestion. Actually, the Exploration of MM-ICL Demonstration Ordering and Exploration of MM-ICL Prompt Construction can reflect how different in-context examples affect the answers. We will add more discussion and add more qualitative analysis for better understanding in the next version.
>
> ---
> **Q3:** As shown in Table 1, in most scenarios, a single textual retriever is sufficient. Therefore, using a multi-modal retriever, which may consume more computational resources, is unnecessary.
>
> **R3:** Thank you for your insightful feedback. This is an interesting research question. Actually, multi-modal retrieval attains better performance in many scenarios like Image Caption and VQA. However, our experiments show that textual retrieval works well for classification and reasoning tasks.
>
> Based on the qualitative analysis, we observe that due to the semantic richness of the labels and rationales, textual retrieval can obtain more similar samples. However, the current multi-modal retrieval struggles with complex text semantics, often favoring image similarity. This aligns with recent work [1,2], which is valuable for future exploration.
>
> In the future, we can consider how to integrate the strengths of text and visual modalities for better performance.
>
> [1] Tong et al. Eyes Wide Shut? Exploring the Visual Shortcomings of Multimodal LLMs. CVPR 2024.
>
> [2] Tong et al. Massproducing failures of multimodal systems with language. NeurIPS 2023.
>
> ---
> **Q4:** Is it appropriate to combine all tasks to provide a universal paradigm for constructing MM-ICL strategies?
>
> **R4:** Thanks for your constructive feedback. We sincerely think that providing a universal paradigm can help researchers conduct unified and fairer comparisons and studies within a unified framework. For example, in each task of this unified paradigm, researchers can explore how to improve a type of task targetedly to achieve better MM-ICL performance. We will add more discussion in the next version.
>
> ---
> **Q5:** Some minor mistakes in the paper.
>
> **R5:** Thanks for your kind mention. We will follow your suggestions to correct these issues one by one (e.g., modifying terminology, highlighting results, and fixing incorrect citation).
>
> ---
> **Q6:** For multi-modal demonstration retrieval, why do you use a multi-modal encoder to calculate the similarity instead of using the product of text and vision single modality similarity?
>
> **R6:** Thanks for your insightful feedback. In our experiment, as shown in Table 1 below, we found that the performance is far inferior to cosine similarity, which is also consistent with our conclusion. We attribute it to the fact that the model is more expected to obtain a semantic direction similarity rather than a distance similarity.
>
> We will add more discussion in the next version.
>
>
> |      | Caption | Caption | VQA | VQA | Classification | Classification | Reasoning | Reasoning | |
> | :--: | :--: | :--: | :--: | :--: | :--: | :--: | :--: | :--: | :--: |
> | | CIDER | BERTScore | Token F1 | BERTScore | Acc | F1 | Accuracy | RAS | AVG |
> | Dot | 10.41 | 86.76 | 18.15 | 84.41 | 58.75 | 40.47 | 52.14 | 91.12 | 55.28 |
> | L2 | 2.58 | 85.3 | 20.85 | 84.67 | 57.98 | 48.95 | 54.96 | 91.5 | 55.85 |
> | Cos | 13.91  | 86.92  | 24.70 | 87.63 | 59.74 | 52.14 | 61.89  | 93.05 | 60.00 |
> ||
>
> Table 1: Performance of different similarity metrics on GPT4o.
>
> ---
> **Q7:** For the OpenAI GPT-4V model, how do you reconstruct the order of the text and the image?
>
> **R7:** Thanks for your valuable feedback. GPT4V API provides a content list that allows for controlling the order of text and images. An example is as follows:
>
> ```json
> {
>   "model": "gpt-4-vision-preview",
>   "messages": [
>     {
>       "role": "user",
>       "content": [
>         {
>           "type": "text",
>           "text": "What’s in this image?"
>         },
>         {
>           "type": "image_url",
>           "image_url": {
>             "url": f"data:image/jpeg;base64,{base64_image}"
>           },
>         }
>         {
>           "type": "text",
>           "text": "What’s in this image?"
>         },
>         {
>           "type": "image_url",
>           "image_url": {
>             "url": f"data:image/jpeg;base64,{base64_image}"
>           },
>
>         }
>       ]
>     }
>   ],
>   "max_tokens": 300
> }
> ```
>
> We will add more details in the next version.

---

> > ### Comment · Reviewer_Pm2y · 2024-08-13
> >
> > Thanks for the authors' detailed response. My concerns have been addressed.

---

> ### Author Response · Authors · 2024-08-13
> **Thanks for Your Time and Effort**
>
> We sincerely appreciate the time and effort you have dedicated to reviewing our response. We are very pleased that all your concerns have been addressed. We sincerely hope that you can consider raising the score after we have addressed all the concerns. We greatly appreciate your selfless contributions to the community.

---

### Author Rebuttal · Authors · 2024-08-05

We thank all reviewers for your insightful and thoughtful feedback.

1. We are greatly encouraged that all reviewers observe that our work addresses **an important research topic** by conducting a thorough exploration of the factors affecting MM-ICL (Reviewer #Pm2y, Reviewer #Wrbn, Reviewer #jbhA, Reviewer #fN5J).
2. We are pleased that reviewers found that our work provides some **insightful and interesting findings**, which can offer **a foundational guide for optimizing MM-ICL strategies in future research** (Reviewer #Pm2y, Reviewer #Wrbn, Reviewer #jbhA).
3. We are also glad that all reviewers found that our work conducts **comprehensive analysis** by exploring 20 strategies across 4 tasks with 6 representative models which is impressive and encouraging (Reviewer #Pm2y, Reviewer #Wrbn, Reviewer #jbhA, Reviewer #fN5J).

We will address all concerns to polish our work according to reviewers’ comments in the next version. Thanks once again for the valuable contributions of all the reviewers.

---

### Decision · Program_Chairs · 2024-09-25

**Decision:**

Accept (poster)

**Comment:**

This submission presented a multimodal framework for In-Context Learning (MMICL). The authors conducted extensive experiments on the three stages of MM-ICL (demonstration retrieval, scheduling and prompting) to explore the factors that affect MMICL performance.
The reviewers raised many questions. The authors were able to answer a number of them. And there was a rich discussion in particular on the type of evaluation carried out to validate the ICL hypotheses, and on the conclusions that are drawn from these “non-standard” Vision-Language evaluations.
In the end, the reviewers were on the whole very convinced. The AC agreed that the strengths in this case outweighed the weaknesses. The final recommendation is to accept. Authors are strongly encouraged to take all comments into account in their final version.